# Pixel-wise Agricultural Image Time Series Classification: Comparisons and a Deformable Prototype-based Approach

## Abstract

Improvements in Earth observation by satellites allow for imagery of ever higher temporal and spatial resolution. Leveraging this data for agricultural monitoring is key for addressing environmental and economic challenges. Current methods for crop segmentation using temporal data either rely on annotated data or are heavily engineered to compensate the lack of supervision. In this paper, we present and compare datasets and methods for both supervised and unsupervised pixel-wise segmentation of satellite image time series (SITS). We also introduce an approach to add invariance to spectral deformations and temporal shifts to classical prototype-based methods such as K-means and Nearest Centroid Classifier (NCC). We study different levels of supervision and show this simple and highly interpretable method achieves the best performance in the low data regime and significantly improves the state of the art for unsupervised classification of agricultural time series on four recent SITS datasets. Our complete code is available at `https://github.com/username/projectname`.

## 1 Introduction

With risks of food supply disruptions, constantly increasing energy needs, population growth and climate change, the threats faced by global agriculture production are plenty (Prosekov & Ivanova, 2018; Mbow et al., 2019). Monitoring crop yield production, controlling plant health and growth, and optimizing crop rotations are among the essential tasks to be carried out at both national and global scales. Because regular ground-based surveys are challenging, remote sensing has very early on appeared as the most practical tool (Justice & Becker-Reshef, 2007).

Thanks to public and commercial satellite launches such as ESA's Sentinel constellation (Drusch et al., 2012; Aschbacher et al., 2017), NASA's Landsat (Woodcock et al., 2008) or Planet's PlanetScope constellation (Boshuizen et al., 2014; Team, 2017), Earth observation is now possible at both high temporal frequency and moderate spatial resolution, typically in the range of 10m/pixel. Sensed data can thus be processed to form satellite image time series (SITS) for further analysis either at the image or pixel level. In particular, several recent agricultural SITS datasets (Kondmann et al., 2021; 2022; Weikmann et al., 2021; Garnot & Landrieu, 2021; Rußwurm et al., 2020) make such data available to the machine learning community, mainly for improving crop type classification.

In this paper, we focus on methods approaching SITS segmentation as multivariate time series classification (MTSC) by considering multi-spectral pixel sequences as the data to classify. While this excludes whole series-based methods like those of Garnot & Landrieu (2021) or Tarasiou et al. (2023) which explicitly leverage the extent of individual parcels, it enables us to extensively evaluate more general MTSC methods that have not yet been applied to agricultural SITS classification. We give particular attention to unsupervised methods as well as interpretability, which we believe would be appealing for extending results beyond well annotated geographical areas.

Our contributions are twofold. First, we benchmark MTSC approaches on four recent SITS datasets (Kondmann et al., 2021; 2022; Weikmann et al., 2021; Garnot & Landrieu, 2021) (Sections 4.1 and 4.3). State-of-the-art supervised methods (Garnot & Landrieu, 2020; Tang et al., 2022; Zhang et al., 2020) are typically complex and require vast amounts of labeled data, i.e., time series with accurate crop labels. We show that,

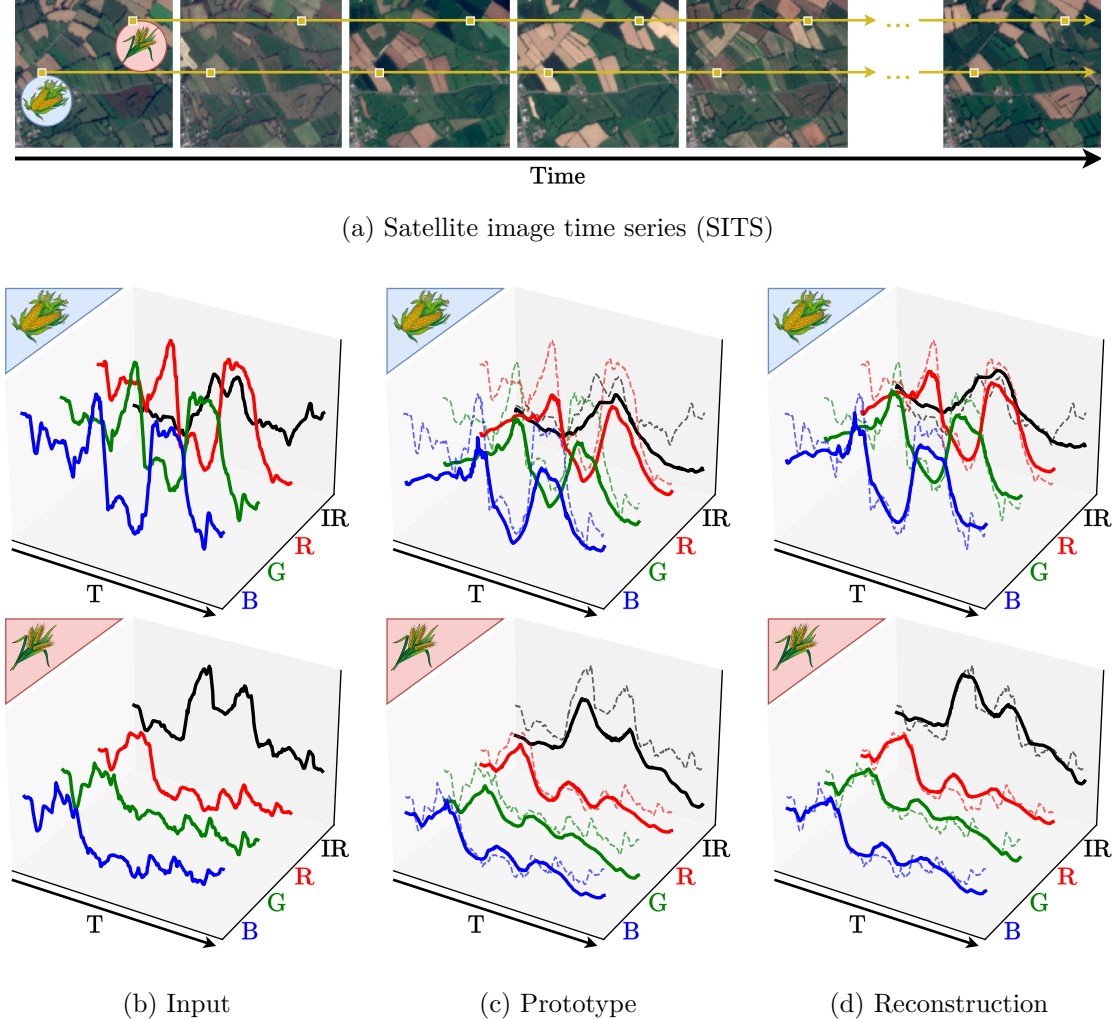

(a) Satellite image time series (SITS)

(b) Input          (c) Prototype          (d) Reconstruction

Figure 1: **Reconstructing pixel sequences from satellite image time series (SITS) through learned prototypes and transformations.** Given a SITS (a), we reconstruct pixel-wise multi-spectral sequences using learned prototypes and transformations. Here, we show the **RGB** and **IR** spectral intensities over time for a corn (🌽) and a wheat (🌾) pixel sequence (b), along with their corresponding prototype before (c) and after (d) transformation.

while they provide strong accuracy boosts over more traditional methods like Random Forest or Support Vector Machine classifiers on datasets with limited domain gap between train and test data, they do not improve over the simple nearest centroid classification baseline on the more challenging DENETHOR (Kondmann et al., 2021) dataset (Section 5.1). In the unsupervised setting, K-means clustering (MacQueen, 1967) and its variant (Petitjean et al., 2011; Zhang et al., 2014) using dynamic time warping (DTW) measure - instead of Euclidean distance - are the strongest baselines (Rivera et al., 2020) (Sections 5.2).

Second, we design a transformation module corresponding to time warping which enables to adapt deep transformation-invariant (DTI) clustering (Monnier et al., 2020) to SITS classification and improve nearest centroid classifier (Cover & Hart, 1967). We refer to our method as DTI-TS. While deep unsupervised methods for SITS classification typically rely either on representation learning or pseudo-labeling, our method learns deformable prototypical sequences (Figure 1) by optimizing a reconstruction loss (Section 3). Our prototypes are learned multivariate time series, typically representing a type of crop, and they can be deformed to model intra-class variabilities. DTI-TS can be trained with or without supervision. In the

unsupervised case, we achieve best scores on all studied datasets by adding spectro-temporal invariance to K-means clustering (MacQueen, 1967). In the supervised case, our model can be seen as an extension of the nearest centroid classifier (Cover & Hart, 1967). In the low data regime, *i.e.* with few labeled image time series, or when there is a temporal domain shift between train and test data, we outperform all competing methods.

## 2 Related Work

We first review methods specifically designed for agricultural SITS classification which are typically supervised and may take as input complete images or individual pixel sequences. When each pixel sequence is considered independently, SITS classification can be seen as a specific case of MTSC, for which both supervised and unsupervised approaches exist, which we review next. Finally, we review transformation-invariant prototype-based classification approaches which we extend to SITS classification in this paper.

**Crop classification with satellite image time series.** Crop classification has historically been achieved at the pixel level, applying traditional machine learning approaches - such as support vector machines or random forests - to vegetation indices like the normalized difference vegetation index (NDVI) (Zheng et al., 2015; Li et al., 2020; Gao et al., 2021). Numerous studies (Kussul et al., 2017; Zhong et al., 2019; Rußwurm & Körner, 2020) now show that in most cases, deep learning methods exhibit superior performance. Deep networks for SITS classification either take individual pixel sequences (Belgiu & Csillik, 2018; Garnot et al., 2020; Garnot & Landrieu, 2020; Blickensdörfer et al., 2022) or series of images (Pelletier et al., 2019; Garnot & Landrieu, 2021; Rußwurm et al., 2023; Mohammadi et al., 2023; Tarasiou et al., 2023) as input. While treating images as a whole may undeniably improve pattern learning for classification as the model can access spatial context information, we focus our work on pixel sequences, which allows us to present a simpler and less restrictive framework that can generalize better to various forms of input data.

**Multivariate times series classification.** Methods achieving MTSC can be divided in two sub-groups: whole series-based techniques and feature-based techniques. Whole series-based methods includes nearest-neighbor search - where the closest neighbor is computed either using Euclidian distance (Cover & Hart, 1967) or DTW (Sakoe & Chiba, 1978; Shokoohi Yekta et al., 2015) - and prototype-based approaches that model a template for each class of the dataset (Seto et al., 2015; Shapira Weber et al., 2019) and classify an input at inference by assigning it to the nearest prototype. Though often simple and intuitive, these methods struggle with in-class temporal distortions or handle them at a high computational cost. Feature-based classifiers include bag-of-patterns methods (Schäfer, 2015; Schäfer & Leser, 2017), shapelet-based techniques (Lines et al., 2012; Bostrom & Bagnall, 2015) and deep encoders like 1D-Convolutionnal Neural Networks (1D-CNNs) (Tang et al., 2022; Ismail Fawaz et al., 2020) or Long Short-Term Memory (LSTM) networks (Ienco et al., 2017; Karim et al., 2019; Zhang et al., 2020). These approaches allow to automatically extract discriminative features from the data, but might be more susceptible to overfitting and tend to be less straightforward to interpret. Instead, our method mixes best of both worlds by learning prototypes along with their deformation as parameters of a deep network in order to efficiently align them with a given input.

**Unsupervised multivariate time series classification.** The classical approach to multivariate time series clustering is to apply K-means (MacQueen, 1967) to the raw time series. This algorithm splits a collection of images into K clusters by jointly optimizing K centroids (centroid step) and the assignment of each data point to the closest centroid (assignment step). DTW has been shown to improve upon K-means for time series clustering in the particular case of SITS (Zhang et al., 2014; Petitjean et al., 2011). DTW is used during both steps of K-means: the centroids are updated as the DTW-barycenter averages of the newly formed clusters and the assignment is performed under DTW. Approaches to multivariate time series clustering often work on improving the representation used by K-means. Methods either extract hand-crafted features (Wang et al., 2005; Rajan & Rayner, 1995; Petitjean et al., 2012) or apply principal component analysis (Li, 2019; Singhal & Seborg, 2005). In Petitjean et al. (2012), mean-shift (Comaniciu & Meer, 2002) is used to segment the image into potential individual crops and K-means features are the means of the spectral bands and the smoothness, area and elongation of the obtained segments. Kalinicheva et al. (2020)

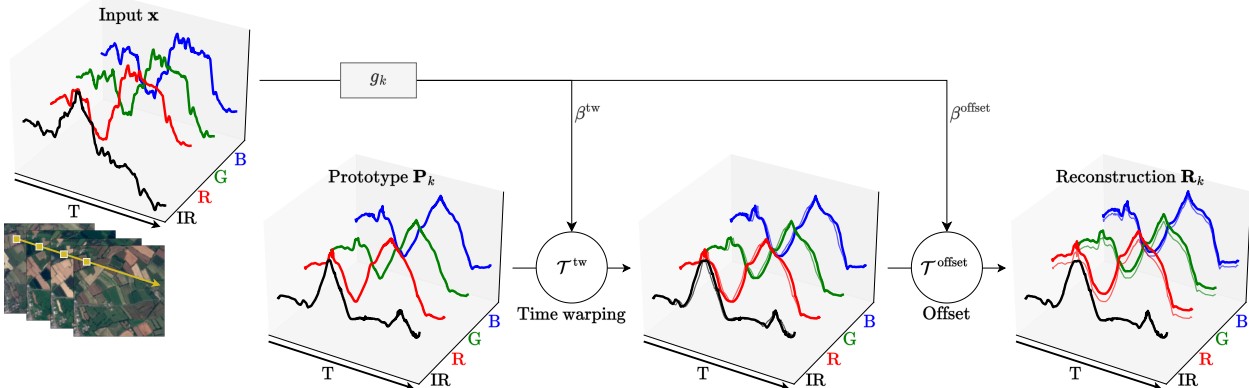

Figure 2: **Overview of DTI-TS.** Our method reconstructs a pixel-wise multi-spectral input sequence, extracted from a SITS, thanks to a prototype to which are successively applied a time warping and an offset. The parameters of these transformations are input-dependent and prototype-specific. The functions $g_{1:K}$ predicting the parameters of the transformations and the prototypes $\mathbf{P}_{1:K}$ can be learned with or without supervision.

reproduce this multi-step scheme but instead (i) applies mean-shift segmentation to a feature map encoded by a 3D spatio-temporal deep convolutional autoencoder, (ii) takes the median of the spectral bands over a segment as a feature representation, and (iii) uses hierarchical clustering to classify each segment. Other deep learning approaches that perform unsupervised classification of time series either use pseudo-labels to train neural networks in a supervised fashion (Guo et al., 2022; Iounousse et al., 2015) or focus on learning deep representations on which clustering can be performed with standards algorithms (Franceschi et al., 2019; Tonekaboni et al., 2021). Deep Temporal Iterative Clustering (DTIC) (Guo et al., 2022) iteratively trains a TempCNN (Pelletier et al., 2019) with pseudo-labeling and performs K-means on the learned features to update the pseudo-labels. Methods that perform deep unsupervised representation learning and clustering simultaneously (Caron et al., 2018; YM. et al., 2020) are promising for time series classification. Although some recent works (Franceschi et al., 2019; Tonekaboni et al., 2021) train supervised classifiers using these learned features on temporal data as input, to the best of our knowledge, no method designed for time series performs classification in a fully unsupervised manner.

**Transformation-invariant prototype-based classification.** The DTI framework (Monnier et al., 2020) jointly learns prototypes and prototype-specific transformations for each sample. The prototypes belong to the input space and their pixel values (in the case of 2D images) or their point coordinates (in the case of 3D point clouds) are free parameters learned while training the model. Each prototype is associated with its own specific transformation network, which predicts transformation parameters for every sample and thus enables the prototype to better reconstruct them. The resulting models can be used for downstream tasks such as classification (Monnier et al., 2020; Loiseau et al., 2022), few-shot segmentation (Loiseau et al., 2021) and multi-object instance discovery (Monnier et al., 2021) and be trained with or without supervision. To the best of our knowledge, the DTI framework has never been applied to the case of time series, for which classifiers need to be invariant to some temporal distortions. Previous works bypass this concern using DTW to compare the samples to classify (Petitjean et al., 2011; Seto et al., 2015) or by applying a transformation field to a selection of control points to distort the time series. Specific to agricultural time series, Nyborg et al. (2022) leverage the fact that temperature is the main factor of temporal variations and uses thermal positional encoding of the temporal dimension to account for temperature change from a year (or location) to another. We use the DTI framework to instead learn the alignment of samples to the prototypes. Shapira Weber et al. (2019) explore a similar idea for generic univariate time series, but, to the best of our knowledge, our paper is the first to perform both supervised and unsupervised transformation-invariant classification for agricultural satellite time series.

## 3 Method

In this section, we explain how we adapt the DTI framework (Monnier et al., 2020) to pixel-wise SITS classification. First, we explain our model and network architecture (Sec. 3.1). Second, we present our training losses in the supervised and unsupervised cases and give implementation and optimization details (Sec. 3.2). We refer to our method as DTI-TS.

**Notation** We use bold letters for multivariate time series (e.g., $\mathbf{a}$, $\mathbf{A}$), brackets [.] to index time series dimensions and we write $a_{1:N}$ for the set $\{a_1, ..., a_n\}$.

### 3.1 Model

**Overview.** An overview of our model is presented in Figure 2. We consider a pixel time series $\mathbf{x}$ in $\mathbb{R}^{T \times C}$ of temporal length $T$ with $C$ spectral bands and we reconstruct it as a transformation of a prototypical time series. We will consider a set of $K$ prototypical time series $\mathbf{P}_{1:K}$, each one being a time series $\mathbf{P}_k \in \mathbb{R}^{T \times C}$ of same size as $\mathbf{x}$ and each intuitively corresponding to a different crop type.

We consider a family of multivariate time series transformations $\mathcal{T}_\beta : \mathbb{R}^{T \times C} \longrightarrow \mathbb{R}^{T \times C}$ parametrized by $\beta$. Our main assumption is that we can faithfully reconstruct the sequence $\mathbf{x}$ by applying to a prototype $\mathbf{P}_k$ a transformation $\mathcal{T}_{g_k(\mathbf{x})}$ with some input-dependent and prototype-specific parameters $g_k(\mathbf{x})$.

We denote by $\mathbf{R}_k(\mathbf{x}) \in \mathbb{R}^{T \times C}$ the reconstruction of the time series $\mathbf{x}$ obtained using a specific prototype $\mathbf{P}_k$ and the prototype-specific parameters $g_k(\mathbf{x})$:

$$\mathbf{R}_k(\mathbf{x}) = \mathcal{T}_{g_k(\mathbf{x})}(\mathbf{P}_k). \tag{1}$$

Intuitively, a prototype corresponds to a type of crop (wheat, oat, etc.) and a given input should be best reconstructed by the prototype of the corresponding class. For this reason, we want the transformations to only account for intra-class variability, which requires defining an adapted transformation model.

**Transformation model.** We have designed a transformation model specific to SITS and based on two transformations: an offset along the spectral dimension and a time warping.

The 'offset' transformation allows the prototypes to be shifted in the spectral dimension to best reconstruct a given input time series (Figure 3a). More formally, the deformation with parameters $\beta^{\text{offset}}$ in $\mathbb{R}^C$ applied to a prototype $\mathbf{P}$ can be written as:

$$\mathcal{T}_{\beta^{\text{offset}}}^{\text{offset}}(\mathbf{P}) = \beta^{\text{offset}} + \mathbf{P}, \tag{2}$$

where the addition is to be understood channel-wise.

The 'time warping' deformation aims at modeling intra-class temporal variability (Figure 3b) and is defined using a thin-plate spline (Bookstein, 1989) transformation along the temporal dimension of the time series. More formally, we start by defining a set of $M$ uniformly spaced landmark time steps $(t_1, ..., t_M)^\top$. Given $M$ target shifts $\beta^{\text{tw}} = (\beta_1^{\text{tw}}, ..., \beta_M^{\text{tw}})^\top$, we denote by $h_{\beta^{\text{tw}}}$ the unique 1D thin-plate spline that maps each $t_m$ to $t'_m = t_m + \beta_m^{\text{tw}}$. Now, given an input pixel time series $\mathbf{x}$ and $\beta^{\text{tw}} \in \mathbb{R}^M$, we define the time warping deformation applied to a prototype $\mathbf{P}$ as:

$$\mathcal{T}_{\beta^{\text{tw}}}^{\text{tw}}(\mathbf{P})[t] = \mathbf{P}[h_{\beta^{\text{tw}}}(t)], \tag{3}$$

for $t \in [1, T]$. Note that the offset is time-independent and that the time warping is channel-independent.

To define our full transformation model, we compose these two transformations, which leads to reconstructions:

$$\mathbf{R}_k(\mathbf{x}) = \mathcal{T}_{\beta^{\text{offset}}}^{\text{offset}} \circ \mathcal{T}_{\beta^{\text{tw}}}^{\text{tw}}(\mathbf{P}_k), \text{ with } (\beta^{\text{offset}}, \beta^{\text{tw}}) = g_k(\mathbf{x}). \tag{4}$$

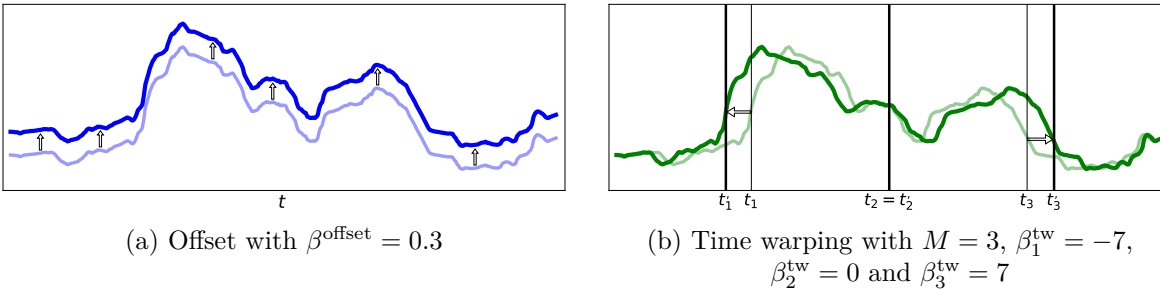

(a) Offset with $\beta^{\text{offset}} = 0.3$

(b) Time warping with $M = 3$, $\beta_1^{\text{tw}} = -7$, $\beta_2^{\text{tw}} = 0$ and $\beta_3^{\text{tw}} = 7$

Figure 3: **Prototype deformations.** We show the visual interpretations of our time series deformations. The offset deformation is time-independent and performed on each spectral band separately. On the other hand, the time warping is channel-independent and achieved by translating landmark time-steps, allowing targeted temporal adjustments.

**Architecture.** The prototypes are multivariate time series whose values in all channels and for all time stamps are free parameters learned through the optimization of a training objective (see Section 3.2). We implement the functions $g_{1:K}$ predicting the transformation parameters as a neural network composed of a shared encoder, for which we use the convolutional network architecture proposed by Wang et al. (2017), and a final linear layer with $K \times (C + M)$ outputs followed by the hyperbolic tangent (tanh) function as activation layer. We interpret this output as $K$ sets of $(C + M)$ parameters for the transformations of the $K$ prototypes. By design, these transformation parameters take values in $[-1, 1]$. This is appropriate for the offset transformation since we normalize the time series before processing, but not for the time warping. We thus multiply the outputs of the network corresponding to the time warping parameters so that the maximum shift of the landmark time steps corresponds to a week. We choose $M$ for each dataset so that we have a landmark time step every month. In the supervised case, we choose $K$ equal to the number of crop classes in each dataset and we set $K$ to 32 in the unsupervised case.

## 3.2 Losses and training

We learn the prototypes $\mathbf{P}_{1:K}$ and the deformation prediction networks $g_{1:K}$ by minimizing a mean loss on a dataset of $N$ multivariate pixel time series $\mathbf{x}_{1:N}$. We define this loss below in the supervised and unsupervised scenarios.

**Unsupervised case.** In this scenario, our loss is composed of two terms. The first one is a reconstruction loss and corresponds to the mean squared error between the input time series and the transformed prototype that best reconstructs it for all pixels $\mathbf{x}$ of the studied dataset:

$$\mathcal{L}_{\text{rec}}(\mathbf{P}_{1:K}, g_{1:K}) = \frac{1}{NTC} \sum_{i=1}^{N} \min_k \left\| \mathbf{x}_i - \mathbf{R}_k(\mathbf{x}_i) \right\|_2^2. \tag{5}$$

The second loss is a regularization term, which prevents high frequencies in the learned prototypes. Indeed, the time warping module allows interpolations between prototype values at consecutive time steps $t$ and $t + 1$, and our network could thus use temporal shifts together with high-frequencies in the prototypes to obtain better reconstructions. To avoid these unwanted high-frequency artifacts, we add a total variation regularization (Rudin et al., 1992):

$$\mathcal{L}_{\text{tv}}(\mathbf{P}_{1:K}) = \frac{1}{K(T-1)C} \sum_{k=1}^{K} \sum_{t=1}^{T-1} \left\| \mathbf{P}_k[t+1] - \mathbf{P}_k[t] \right\|_2. \tag{6}$$

The full training loss without supervision is thus:

$$\mathcal{L}_{\text{unsup}}(\mathbf{P}_{1:K}, g_{1:K}) = \mathcal{L}_{\text{rec}}(\mathbf{P}_{1:K}, g_{1:K}) + \lambda \mathcal{L}_{\text{tv}}(\mathbf{P}_{1:K}), \tag{7}$$

with $\lambda$ a scalar hyperparameter set to 1 in all our experiments.

**Supervised case.** In the supervised scenario, we choose $K$ as the true number of classes in the studied dataset, and set a one-to-one correspondence between each prototype and one class. We leverage this knowledge of the class labels to define two losses. Let $y_i \in \{1, ..., K\}$ be the class label of input pixel $\mathbf{x}_i$. First, a reconstruction loss similar to (5) penalizes the mean squared error between an input and its reconstruction using the true-class prototype:

$$\mathcal{L}_{\text{rec\_sup}}(\mathbf{P}_{1:K}, g_{1:K}) = \frac{1}{NTC} \sum_{i=1}^{N} \left\| \mathbf{x}_i - \mathbf{R}_{y_i}(\mathbf{x}) \right\|_2^2. \tag{8}$$

Second, in order to boost the discriminative power of our model, we add a contrastive loss (Loiseau et al., 2022) based on the reconstruction error:

$$\mathcal{L}_{\text{cont}}(\mathbf{P}_{1:K}, g_{1:K}) = -\frac{1}{N} \sum_{i=1}^{N} \log \left( \frac{\exp \left( - \left\| \mathbf{x}_i - \mathbf{R}_{y_i}(\mathbf{x}) \right\|_2^2 \right)}{\sum_{k=1}^{K} \exp \left( - \left\| \mathbf{x}_i - \mathbf{R}_k(\mathbf{x}) \right\|_2^2 \right)} \right). \tag{9}$$

We also use the same total variation regularization as in the unsupervised case, and the full training loss under supervision is:

$$\mathcal{L}_{\text{sup}}(\mathbf{P}_{1:K}, g_{1:K}) = \mathcal{L}_{\text{rec\_sup}}(\mathbf{P}_{1:K}, g_{1:K}) + \mu \mathcal{L}_{\text{tv}}(\mathbf{P}_{1:K}) + \nu \mathcal{L}_{\text{cont}}(\mathbf{P}_{1:K}, g_{1:K}), \tag{10}$$

with $\mu$ and $\nu$ two hyperparameters equal to 1 and 0.01 respectively in all our experiments.

**Initialization.** The learnable parameters of our model are (i) the prototypes, (ii) the encoder and (iii) the time warping and offset decoders. We initialize our prototypes with the centroids learned by NCC (resp. K-means) in the supervised (resp. unsupervised) case. Default Kaming He initialization (He et al., 2015) is used for the encoder while the parameters of both decoders are set to zero. This ensures that at initialization the predicted transformations are the identity.

**Optimization.** Parameters are learned using the ADAM optimizer (Kingma & Ba, 2015) with a learning rate of $10^{-5}$. We train our model following a curriculum modeling scheme (Elman, 1993; Monnier et al., 2020): we progressively increase the model complexity by first training without deformation, then adding the time warp deformation and finally the offset deformation. We add transformations when the mean accuracy does not increase in the supervised setting and, in the unsupervised setting, when the reconstruction loss does not decrease, after 5 validation steps. Note that the contrastive loss is only activated at the end of the curriculum in the supervised setting.

### 3.3 Handling missing data

Our method, as presented in Section 3, is designed for uniformly sampled constant-sized time series. While satellite time series from PlanetScope are pre-processed to obtain such regular data, time series acquired by Sentinel 2 have at most a data point every 5 days due to a lower revisit frequency, and additional missing dates because of clouds or shadows. To handle such non-regularly sampled time series, the remote sensing literature proposes several gap filling methods (Belda et al., 2020; Julien & Sobrino, 2019; Kandasamy et al., 2013). Instead, since our method is distance-based, we propose (i) to filter the input data to prevent possible outliers and (ii) to only compare inputs and prototypes on time stamps for which the input is defined.

Let us consider a specific time series, acquired over a period of length $T$ but with missing data. We define the associated raw time series $\mathbf{x}_{\text{raw}} \in \mathbb{R}^{T \times C}$ by setting zero values for missing time stamps and the associated binary mask $\mathbf{m}_{\text{raw}} \in \{0,1\}^T$, equal to 0 for missing time stamps and 1 otherwise. We define the filtered time series $\mathbf{x}$ extracted from $\mathbf{x}_{\text{raw}}$ and $\mathbf{m}_{\text{raw}}$ through Gaussian filtering for $t \in [1, T]$ by:

$$\mathbf{x}[t] = \frac{1}{\mathbf{m}[t]} \sum_{t'=1}^{T} \mathcal{G}_{t,\sigma}[t'] \cdot \mathbf{x}_{\text{raw}}[t'], \tag{11}$$

Table 1: **Comparison of studied datasets.** The datasets we study cover different regions (France, Austria, South Africa and Germany). We distinguish between datasets where train and test splits differ only spatially (Spat.) and where they differ both spatially and temporally (Spat. & Temp.). Time series can have daily data (✔) or missing data (✗). Additionally, we report the length of the time series $T$, the number of spectral bands $C$ and the number of classes $K$. The last column shows the split sizes as train | val | test, except for PASTIS where we follow the 5-fold procedure described in Garnot & Landrieu (2021) and we show the size of each of the folds.

| Dataset | Country | $T$ | $C$ | $K$ | Train/Test shift | Satellite(s) | Daily | Split size (x $10^6$) |
|---|---|---|---|---|---|---|---|---|
| PASTIS | 🇫🇷 | 406 | 10 | 19 | Spat. | Sentinel 2 | ✗ | 7.3 \| 7.3 \| 7.3 \| 7.0 \| 7.1 |
| TimeSen2Crop | 🇦🇹 | 363 | 9 | 16 | Spat. | Sentinel 2 | ✗ | 0.8 \| 0.1 \| 0.1 |
| SA | 🇿🇦 | 244 | 4 | 5 | Spat. | PlanetScope | ✔ | 60.1 \| 10.1 \| 32.0 |
| DENETHOR | 🇩🇪 | 365 | 4 | 9 | Spat. & Temp. | PlanetScope | ✔ | 20.6 \| 3.2 \| 22.8 |

with

$$\mathcal{G}_{t,\sigma}[t'] = \exp\left(-\frac{(t'-t)^2}{2\sigma^2}\right), \tag{12}$$

where $\sigma$ is a hyperparameter set to 7 days in our experiments. We also define the associated filtered mask $\mathbf{m}$ for $t \in [1, T]$ by:

$$\mathbf{m}[t] = \sum_{t'=1}^{T} \mathcal{G}_{t,\sigma}[t'] \cdot \mathbf{m}_{\text{raw}}[t'], \tag{13}$$

for $t \in [1, T]$ and with the same hyperparameter $\sigma$.

Using directly this filtered time series to compute our mean square errors would lead to large errors, because data might be missing for long time periods. Thus, we modify the losses $\mathcal{L}_{\text{rec}}$ and $\mathcal{L}_{\text{rec\_sup}}$ by replacing the reconstruction error between a time series $\mathbf{x}$ and reconstruction $\mathbf{R}$,

$$\frac{1}{TC}\left\|\mathbf{x} - \mathbf{R}\right\|_2^2 = \frac{1}{C}\sum_{t=1}^{T}\frac{1}{T}\left\|\mathbf{x}[t] - \mathbf{R}[t]\right\|_2^2, \tag{14}$$

in Equations (5) and (8) by a weighted mean squared error:

$$\frac{1}{C}\sum_{t=1}^{T}\frac{\mathbf{m}[t]}{\sum_{t'=1}^{T}\mathbf{m}[t']}\left\|\mathbf{x}[t] - \mathbf{R}[t]\right\|_2^2. \tag{15}$$

This adapted loss gives more weight to time stamps $t$ corresponding to true data acquisitions.

In Appendix A, we justify these design choices and demonstrate that they result in superior performance when compared to alternative standard filtering schemes, both for our method and NCC.

## 4 Experiments

### 4.1 Datasets

We consider four recent open-source datasets on which we evaluate our method and multiple baselines. Details about these datasets can be found in Table 1.

**PASTIS (Garnot & Landrieu, 2021).** This dataset contains Sentinel-2 satellite patches within the French metropolitan area, acquired from September 1, 2018 to October 31, 2019. Each image time series contains a variable number of images that can show clouds and/or shadows. We pre-process the dataset and remove most of the cloudy/shadowy pixels using a classical thresholding approach on the blue reflectance (Breon & Colzy, 1999). We consider each of the pixels of the 2433 $128 \times 128$ image time series as

independent time series, except those corresponding to the 'void' class, leading to 36M times series. Each is labeled with one of 19 classes (including a *background*, i.e., non-agricultural class). We follow the same 5-fold evaluation procedure as described in Garnot & Landrieu (2021), with at least 1km separating images from different folds to ensure distinct spatial coverage between them.

**TimeSen2Crop (Weikmann et al., 2021).** This dataset is also built from Sentinel-2 satellite images, but covering Austrian agricultural parcels and acquired between September 3, 2017 and September 1, 2018. It does not provide images but directly 1M pixel time series of variable lengths. We pre-process these time series by removing the time-stamps associated to the 'shadow' and 'clouds' annotations provided in the dataset. Each time series is labeled with one of 16 types of crops. We follow the same train/val/test splitting as in Weikmann et al. (2021) where each split covers a different area in Austria.

**SA (Kondmann et al., 2022).** This dataset is built from images from the PlanetScope constellation of Cubesats satellite covering agricultural areas in South Africa, and contains daily time series from April 1, 2017 to November 31, 2017. Acquisitions are fused using Planet Fusion[1] to compensate for possible missing dates, clouds or shadows so that the provided data consists in clean daily image time series. The dataset contains 4151 single-field images time series from which we extract 102M pixel time series. Each time series is labeled with one of 5 types of crops. We keep the same train/test splitting of the data and reserve 15% of the train set for validation purposes. We make sure that the obtained train and validation set do not have pixel time series extracted from the same field image.

**DENETHOR (Kondmann et al., 2021).** This dataset is also built from Cubesats images but covers agricultural areas in Germany. The training set is built from daily time series acquired from January 1, 2018 to December 31, 2018, while the test set is built from time series acquired from January 1, 2019 to December 31, 2019. Similar to SA, the dataset has been pre-processed to provide clean daily time series. It contains 4561 single-field images time series from which we extract 47M independent pixel time series. Each time series is labeled with one of 9 types of crops. Again, we use the original splits of the data, with 15% of the training set kept for validation. All splits cover distinct areas in Germany.

The time shift between train and test sets makes DENETHOR significantly more challenging than the three other datasets. On Figure 4, we illustrate this domain gap by showing the mean NDVI of three random classes for each dataset on the train and test splits. The train and test curves are more dissimilar with DENETHOR than any other dataset, and significant differences in the NDVI curves remain after alignment using our time warping and offset transformation.

## 4.2 Metrics

We provide two metrics for evaluating classification accuracy: overall accuracy (OA) and mean accuracy (MA). OA is computed as the ratio of correct and total predictions:

$$OA = \frac{TP + TN}{TP + TN + FP + FN}, \tag{16}$$

where TP, TN, FP and FN correspond to true positive, true negative, false positive and false negative, respectively. MA is the class-averaged classification accuracy:

$$MA = \frac{1}{K} \sum_{k=1}^{K} OA(\{\mathbf{x}_i | y_i = k\}). \tag{17}$$

It is important to note that the datasets under consideration show a high degree of imbalance, making MA a more appropriate and informative metric for evaluating classification performance. For this reason, OA scores are shown in gray in Tables 2, 3 and 4.

---

[1]https://assets.planet.com/docs/Fusion-Tech-Spec__v1.0.0.pdf

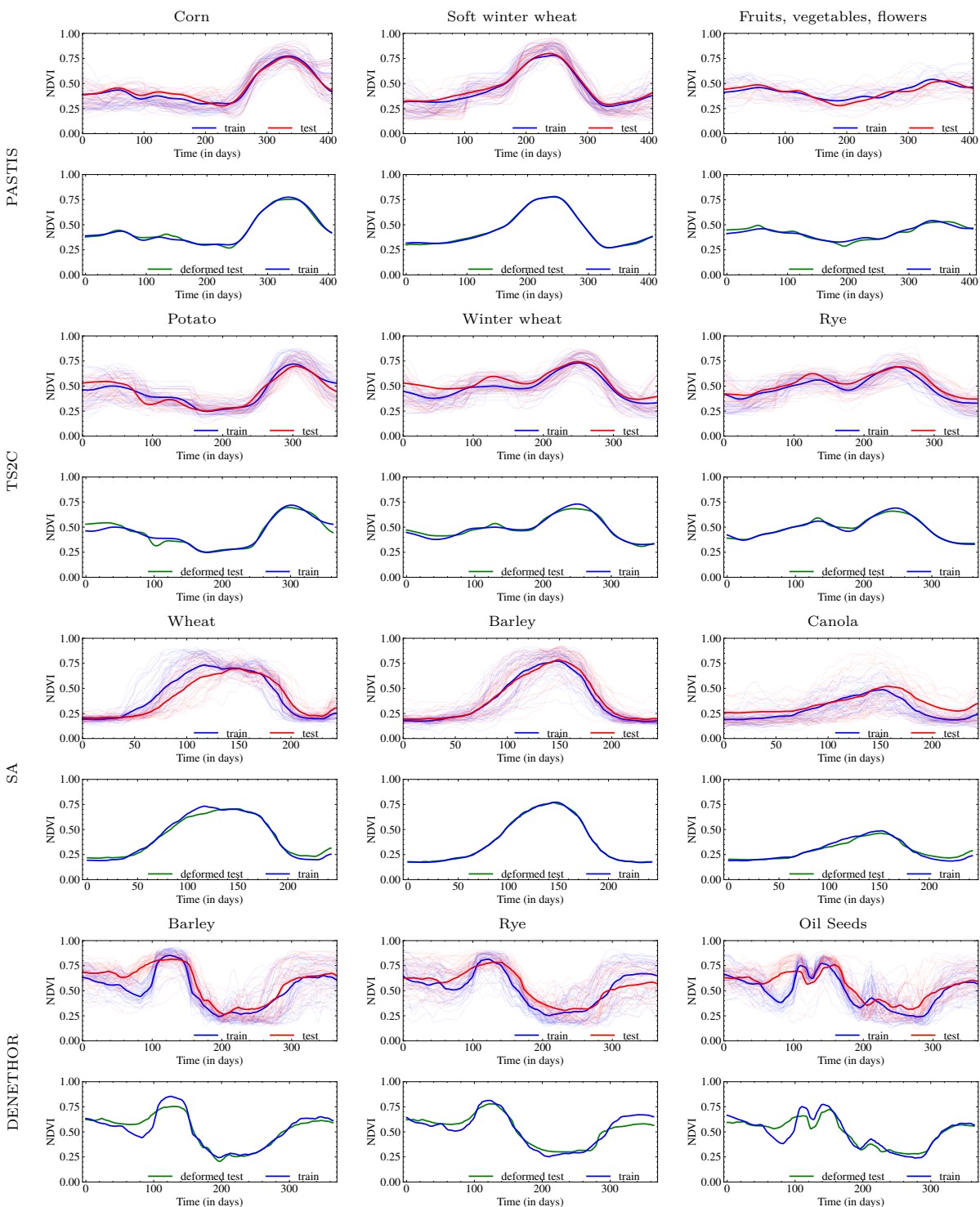

Figure 4: **Temporal domain gap and alignment.** For each dataset, we show the mean NDVI of three randomly selected classes on train and test splits (top row). Then we align the test curve to the train mean NDVI by optimizing the parameters of the time warping and the offset transformations with gradient descent (bottom row).

### 4.3 Baselines

We validate our approach in two settings: supervised (classification) and unsupervised (clustering). For each setting, we describe below the methods evaluated in this work.

#### 4.3.1 Time series classification

The purpose of this section is to benchmark classic and state-of-the-art MTSC methods for crop classification in SITS data:

- **NCC (Duda et al., 1973).** The nearest centroid classifier (NCC) assigns to a test sample the label of the closest class average time series using the Euclidean distance. We also report the extension of NCC with our method to add invariance to time warping and sequence offset, as well as adding our contrastive loss.

- **1NN (Cover & Hart, 1967) and 1NN-DTW (Seto et al., 2015).** The first nearest neighbor algorithm assigns to a test sample the label of its closest neighbor in the train set, with respect to a given distance. This algorithm is computationally costly and, since the datasets under study typically contain millions of pixel time series, we search for neighbors of test samples in a random 0.1% subset of the train set and report the average over 5 runs with different subsets. We evaluate the nearest neighbor algorithm using the Euclidean distance (1NN) as well as using the dynamic time warping (1NN-DTW) measure on the TimeSen2Crop dataset which is small enough to compute it in a reasonable time.

- **SVM (Cortes & Vapnik, 1995).** We trained a linear support vector machine (SVM) in the input space using scikit-learn library (Pedregosa et al., 2011).

- **Random Forest (Ho, 1995).** We evaluated the performance of a Random Forest of a hundred trees built in the input space using scikit-learn library Pedregosa et al. (2011).

- **MLSTM-FCN (Karim et al., 2019).** MLSTM-FCN is a two-branch neural network concatenating the ouputs of an LSTM and a 1D-CNN to better encode time series. We use a non-official PyTorch implementation[2] of MLSTM-FCN.

- **TapNet (Zhang et al., 2020).** TapNet uses a similar architecture to MLST-FCN to learn a low-dimensional representation of the data. Additionally, Zhang et al. (2020) learn class prototypes in this latent space using the softmin of the euclidian distances of the embedding to the different class prototypes as classification scores. The official PyTorch implementation[3] is designed for datasets with a size range from 27 to 10,992 only, while our datasets contain millions of time series. Thus, based on the official implementation, we implemented a batch version of TapNet which we use for our experiments.

- **OS-CNN (Tang et al., 2022).** The Omni-Scale CNN is a 1D convolutional neural network that has shown ability to robustly capture the best time scale because it covers all the receptive field sizes in an efficient manner. We use the official implementation[4] with default parameters.

- **MLP+LTAE (Garnot & Landrieu, 2020).** The Lightweight Temporal Attention Encoder (LTAE) is an attention-based network. Used along with a Pixel Set Encoder (PSE) (Garnot et al., 2020), LTAE achieves good performances on images. To adapt it to time series, we instead use a MLP as encoder. We refer to this method as MLP+LTAE and we use the official PyTorch implementation[5] of LTAE.

---

[2]github.com/timeseriesAI/tsai
[3]github.com/kdd2019-tapnet/tapnet
[4]github.com/Wensi-Tang/OS-CNN
[5]github.com/VSainteuf/lightweight-temporal-attention-pytorch

- **UTAE (Garnot et al., 2020).** In addition to SITS methods, we also report the scores of U-net with Temporal Attention Encoder (UTAE) on PASTIS dataset. This method leverages complete (constant-size) images. Since it can learn from the spatial context of a given pixel, this state-of-the-art image sequence segmentation approach is expected to perform better than pixel-based MTSC approaches and is reported for reference.

### 4.3.2  Time series clustering

In the unsupervised setting, we compare our method to other clustering approaches applied on learned features or directly on the time series:

- **K-means (Bottou & Bengio, 1994).** We apply the classic K-means algorithm on the multivariate pixel time series directly. Clustering is performed on all splits (train, val and test). Then we determine the most frequently occurring class in each cluster, considering training data only. The result is used as label for the entire cluster. We use the gradient descent version (Bottou & Bengio, 1994) K-means with empty cluster reassignment (Caron et al., 2018; Monnier et al., 2020).

- **K-means-DTW (Petitjean et al., 2011).** The K-means algorithm is applied in this case with a dynamic time warping measure instead of the usual Euclidean distance. To this end, we use the differentiable Soft-DTW (Cuturi & Blondel, 2017) version of DTW and its Pytorch implementation (Maghoumi et al., 2021).

- **USRL (Franceschi et al., 2019) + K-means.** USRL is an encoder trained in an unsupervised manner to represent time series by a 320-dimensional vector. We train USRL on all splits of each dataset, then apply K-means in the feature space. We use the official implementation[6] of USRL with default parameters.

- **DTAN (Shapira Weber et al., 2019) + K-means.** DTAN is an unsupervised method for aligning temporally all the time series of a given set. K-means is applied on data from all splits after alignment with DTAN. We use the official implementation[7] of DTAN with default parameters.

We evaluate all methods with $K = 32$ clusters. We discuss this choice in Appendix B.

## 5  Results and Discussion

In this section, we first compare our method to top-performing supervised methods proposed in the literature for MTSC as well as traditional machine learning methods (Sec. 5.1). We then demonstrate that our method outperforms the K-means baseline on all four datasets (Sec. 5.2) thanks to the design choices for our time series deformations. Finally, we discuss qualitative results and the interpretability of out method (Sec. 5.3).

### 5.1  Time series classification

Results on the DENETHOR dataset are qualitatively very different from the results on the other datasets. We believe this is because DENETHOR has train and test splits corresponding to two distinct years. We thus analyze it separately.

**Results on PASTIS, TimeSen2Crop and SA.**  As expected, since UTAE can leverage knowledge on the spatial context of each pixel, it achieves the best score on PASTIS dataset by +2.0% in OA and +5.5% in MA. Our improvements over the NCC method (Duda et al., 1973) - adding time warping deformation, offset deformations and contrastive loss (9) - consistently boost the mean accuracy. The improvement obtained by adding transformation modeling comes from a better capability to model the data, as confirmed by the detailed results reported in the left part of Table 4, where one can see the reconstruction error (i.e.

---

[6]github.com/White-Link/UnsupervisedScalableRepresentationLearningTimeSeries
[7]github.com/BGU-CS-VIL/dtan

Table 2: **Performance comparison for classification on all datasets.** We report for our method and competing methods, the number of trainable parameters (#param) when trained on PASTIS, the overall accuracy (OA) and the mean class accuracy (MA). We distinguish with a background color the DENETHOR dataset - where train and test splits are acquired during different periods - from the others. 1NN-DTW is tested on TimeSen2Crop dataset only, due to the expensive cost of the algorithm. We separate results in 3 parts: the image level method UTAE, MTSC methods and different ablations of DTI-TS. We put in bold the best method in each of the 3 parts and underline the absolute best for each dataset. We report the average inference time of each method to process a batch of 2,048 time series from TS2C on a single NVIDIA GeForce RTX 2080 Ti GPU.

| Method | #param (x1000) | Inf. time (ms/batch) | PASTIS OA↑ | PASTIS MA↑ | TS2C OA↑ | TS2C MA↑ | SA OA↑ | SA MA↑ | DENETH. OA↑ | DENETH. MA↑ |
|---|---|---|---|---|---|---|---|---|---|---|
| UTAE (Garnot & Landrieu, 2021) | 1 087 | — | 83.3 | 73.6 | — | — | — | — | — | — |
| MLP + LTAE (Garnot & Landrieu, 2020) | 320 | 78 | 80.6 | 65.9 | 88.7 | 80.9 | 67.4 | 63.7 | 55.6 | 43.6 |
| OS-CNN (Tang et al., 2022) | 4 729 | 119 | 81.3 | 68.1 | 87.9 | 81.2 | 64.6 | 60.3 | 49.0 | 39.2 |
| TapNet (Zhang et al., 2020) | 1 882 | 229 | 78.0 | 60.3 | 83.1 | 77.3 | 59.6 | 56.7 | 53.1 | 43.7 |
| MLSTM-FCN (Karim et al., 2019) | 490 | 11 | 44.4 | 10.9 | 58.7 | 44.0 | 56.1 | 47.9 | 58.2 | 48.3 |
| SVM (Cortes & Vapnik, 1995) | 77 | 48 | 76.3 | 48.7 | 74.9 | 56.1 | 64.6 | 52.8 | 35.6 | 28.6 |
| Random Forest (Ho, 1995) | 16 | 140 | 76.6 | 46.6 | 66.9 | 50.2 | 69.9 | 61.3 | 59.9 | 51.6 |
| 1NN-DTW (Seto et al., 2015) | 0 | >10⁴ | — | — | 32.2 | 23.0 | — | — | — | — |
| 1NN (Cover & Hart, 1967) | 0 | 6 | 65.8 | 40.1 | 43.9 | 35.0 | 60.7 | 54.9 | 56.7 | 48.2 |
| NCC (Duda et al., 1973) | 77 | 24 | 56.5 | 48.4 | 57.1 | 49.9 | 51.3 | 46.4 | 61.3 | 55.5 |
| DTI-TS: NCC + time warping | 398 | 97 | 56.2 | 51.4 | 59.9 | 52.3 | 54.5 | 49.7 | 62.4 | 56.4 |
| + offset | 423 | 97 | 53.5 | 53.8 | 57.3 | 55.0 | 60.6 | 50.0 | 59.8 | 62.9 |
| + contrastive loss | 423 | 97 | 73.7 | 59.1 | 78.5 | 70.5 | 62.3 | 54.9 | 56.5 | 54.2 |

$\mathcal{L}_{rec}$) significantly decreases when adding these transformations. Note that on the contrary, adding the discriminative loss increase the accuracy at the cost of decreasing the quality of the reconstruction error. Our complete supervised approach outperforms both the nearest neighbor based methods and MLSTM-FCN. However, it is still significantly outperformed by top MTSC methods. This is not surprising, since these methods are able to learn complex embeddings that capture subtle signal variations, e.g. thanks to a temporal attention mechanism (Garnot & Landrieu, 2020) or to multiple-sized receptive fields (Tang et al., 2022). Note however that in doing so, they loose the interpretability of simpler approaches such as 1NN or NCC, which our method is designed to keep.

**Results on DENETHOR.** Because the data we use is highly dependent on weather conditions, subsets acquired on distinct years follow significantly different distributions (Kondmann et al., 2021). Because of their complexity, other methods struggle to deal with this domain shift. In this setting, our extension of NCC to incorporate specific meaningful deformations achieves better performances than all the other MTSC methods we evaluated. However, adding the contrastive loss significantly degrades the results. We believe this is again due to the temporal domain shift between train and test data. This analysis is supported by results reported in Table 4 which show that on the validation set of DENETHOR, which is sampled from the same year as the training data, adding the constrative loss significantly boost the results, similar to the other dataset. One can also see again on DENETHOR the benefits of modeling the deformations in terms of reconstruction error.

**Low data regime.** Our method is also beneficial when only few annotated images are available at training time. In Figure 5, we plot the MA obtained by NCC, MLP+LTAE, OS-CNN, TapNet and our method depending on the proportion of the SITS of PASTIS dataset. While all methods benefit from more training data, our prototype-based approach generalizes better from few annotated samples. When using 4% of the dataset or less, *i.e.* 60 annotated image time series or less, our method is the best of all MTSC methods benchmarked in this paper. Training on 1% of the data it outperforms MLP+LTAE by +4.7% in MA, TapNet by +8.8% and OS-CNN by +10.9% but is not able to clearly do better than the NCC baseline. Using 2% or

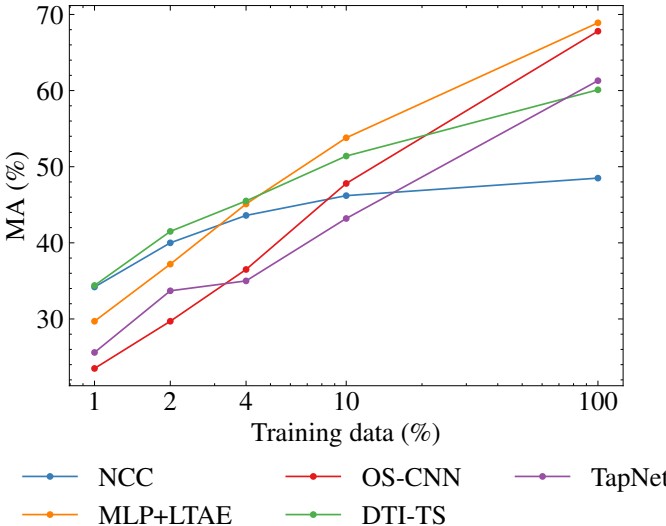

Figure 5: **Low data regime on PASTIS dataset.** Using Fold 2 of PASTIS dataset, we train on only 1, 2, 4 or 10% of the image time series of the training set. For the 1%, 2% and 4% samples, we show the average for 5 random different subsets.

Table 3: **Performance comparison for clustering on all datasets.** We report for our method and competing methods the number of trainable parameters (#param) when trained on PASTIS, the accuracy (OA) and the mean class accuracy (MA). K-means clustering is run with 32 clusters for all methods for fair comparison. We distinguish with a background color the DENETHOR dataset - where train and test splits are acquired during different periods - from the others. K-means-DTW is tested on TimeSen2Crop dataset only, due to the expensive cost of the algorithm. We report the average inference time of each method to process a batch of 2,048 time series from TS2C on a single NVIDIA GeForce RTX 2080 Ti GPU.

| | #param | Inf. time | PASTIS | | TS2C | | SA | | DENETH. | |
| Method | (x1000) | (ms/batch) | OA↑ | MA↑ | OA↑ | MA↑ | OA↑ | MA↑ | OA↑ | MA↑ |
|---|---|---|---|---|---|---|---|---|---|---|
| K-means-DTW (Petitjean et al., 2011) | 130 | $>10^4$ | — | — | 40.5 | 26.8 | — | — | — | — |
| USRL (Franceschi et al., 2019)+K-means | 259 | 193 | 63.9 | 20.4 | 34.9 | 23.6 | 60.9 | 48.6 | 54.0 | 46.4 |
| DTAN (Shapira Weber et al., 2019)+K-means | 256 | 28 | 65.6 | 21.4 | 47.7 | 29.3 | 60.5 | 48.6 | 46.3 | 36.9 |
| K-means (Bottou & Bengio, 1994) | 130 | 7 | 69.0 | 29.8 | 49.5 | 32.5 | 61.9 | 47.8 | 57.2 | 48.5 |
| DTI-TS: K-means + time warping | 471 | 13 | **69.1** | **30.4** | **52.3** | **36.0** | **64.1** | **51.7** | 57.6 | 51.1 |
| + offset | 512 | 18 | 67.7 | 28.6 | 52.0 | 35.5 | 63.6 | 50.4 | **58.5** | **52.6** |

4% of the dataset, DTI-TS clearly improves over NCC and still has better scores than MLP+LTAE, TapNet and OS-CNN.

## 5.2 Time series clustering

In this section, we demonstrate clear boosts provided by our method on the four SITS datasets we study. Our method outperforms all the other baselines on the four datasets, always achieving the best mean accuracy. In particular, our time warping transformation appears to be the best way to handle temporal information when clustering agricultural time series. Indeed, DTAN+K-means leads to a significantly less accurate clustering than simple K-means. It confirms that temporal information is crucial when clustering agricultural time series: when DTAN aligns temporally all the sequences of a given dataset, it probably discards discriminative information, leading to poor performance. The same conclusion can be drawn from the results of K-means-DTW on TimeSen2Crop. In contrast, our time warping appears as constrained enough to both reach satisfying scores and account for the temporal diversity of the data.

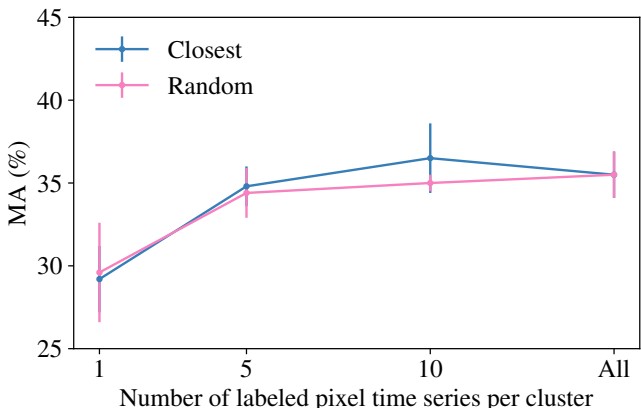

Figure 6: **Number of labeled pixel times series used to assign prototypes.** We label each of the $K = 32$ prototypes obtained on TS2C training set using the 1, 5 and 10 closest - or 1, 5 and 10 random - pixel time series in its cluster on the training set. We report the mean accuracy averaged over 5 runs and compare it to when using all annotated pixel time series of the training set.

Using an offset transformation on the spectral intensities consistently results in improved sample reconstruction using our prototypes, as demonstrated in Table 4. However, it only increases classification scores for DENETHOR. We attribute this improvement to the offset transformation's ability to better handle the domain shift between the training and testing data on the DENETHOR dataset. The results on the other datasets suggest that this transformation accounts for more than just intra-class variability, leading to less accurate classification scores, as discussed in Section 5.4. For all the methods compared above, we label the clusters with the most frequently occurring class in each of them on the train set. This can correspond to millions of annotated pixel time series being used, but our method works with far less annotations. We report in Figure 6 the MA of our method on TS2C when only 1, 5 or 10 annotated pixel time series used to decide for each cluster's label. We sample either random time series in each cluster ('Random') or select the time series that are best reconstructed by the given prototype ('Closest'). There is a clear 5% performance drop when a single time series is used to label each cluster. However, using 5 time series per cluster is already enough to recover scores similar to the ones obtained using the full training dataset. For TS2C, this amounts to 0.001% of all the training data.

## 5.3 Qualitative evaluation

### 5.3.1 Land cover maps

We provide in Figure 7 a visualization of the land cover maps obtained by our method and competing supervised approaches on PASTIS dataset. We show 4 randomly selected image time series from Fold 2 test set. One can see how our method improves over NCC by allowing pixels of the same field to be classified similarly. We highlight with black circles (◯) examples of areas where NCC gives different labels to central and border pixels whereas our method use the same deformable prototype to reconstruct all pixels of the field. OS-CNN and TapNet fail to classify properly the sea as background which we highlight with a yellow circle (◯). TapNet land cover maps are the most noisy, with a salt-and-pepper effect that is particularly noticeable on the third row. MLP+LTAE is the best at maintaining spatial consistency within crop classes and at accurately delineating boundaries. Similarly, we visually compare in Figure 8 land cover maps obtained with K-means and our method. We highlight with black circles (◯) areas where our approach distinguish more faithfully agricultural parcels from the background class than K-means. Since cluster labeling is performed through majority voting, most clusters get assign to the majority background class on PASTIS: it is the case for 47% of K-means clusters on Fold 2. However, our deformable prototypes can represent the same class with less clusters, hence only 41% of them account for the background class. All pixel-wise SITS semantic

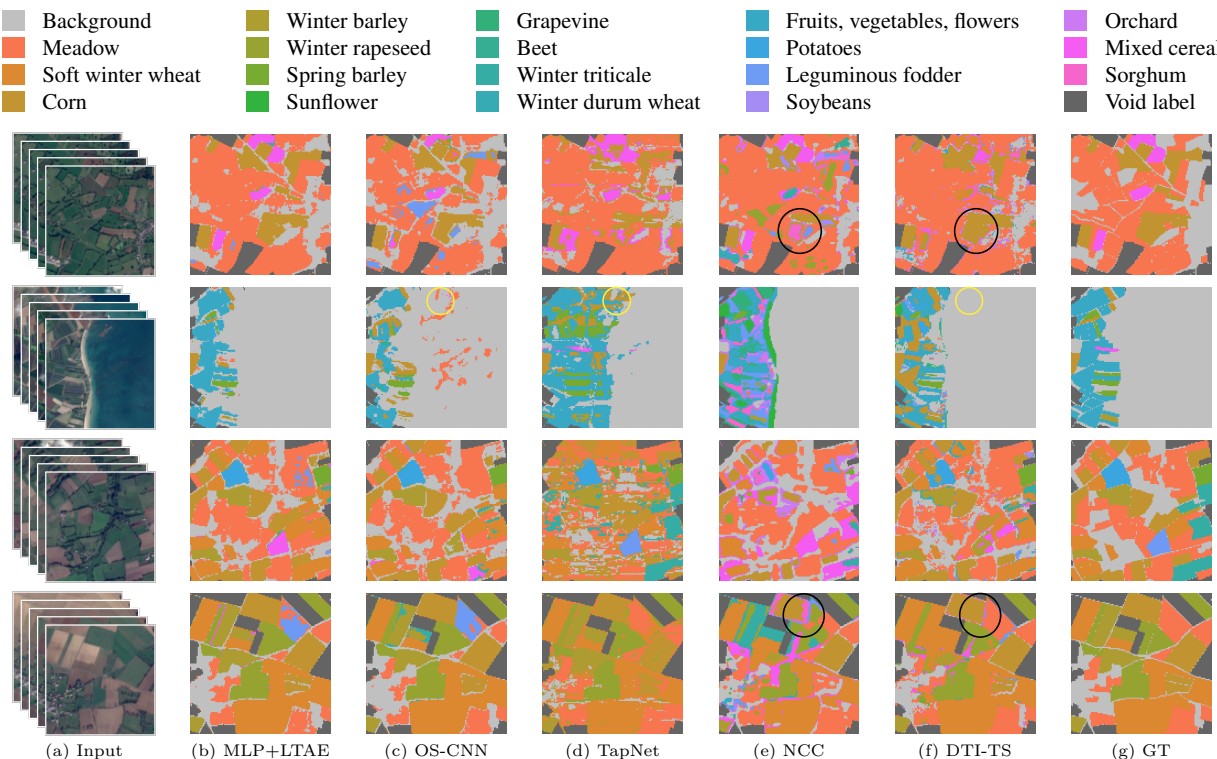

Figure 7: **Qualitative comparisons of supervised methods.** We show predicted segmentation maps for best-performing supervised methods (b-d), NCC (e) and our method (f) for randomly selected SITS from Fold 2 test set of PASTIS (a). Dark grey segments correspond to the *void* class and are ignored by all methods. The legend above is used for all other semantic segmentation visualizations of this paper.

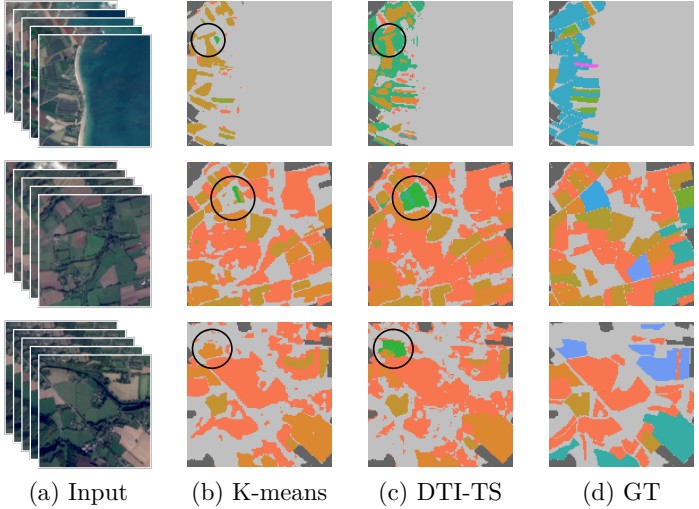

Figure 8: **Qualitative comparisons of unsupervised methods.** We show predicted segmentation maps for K-means (b) and our method (c) for randomly selected SITS from Fold 2 test set of PASTIS (a). Dark grey segments correspond to the *void* class and are ignored by all methods.

segmentation methods can benefit from a post-processing step taking into account spatial information, *e.g.*, aggregating predictions in nearby pixels, or on each field. While such post-processing is not the focus of our paper, we demonstrate in Appendix C the benefits of several such post-processing methods.

Table 4: **Detailed evaluation of our method.** We show the impact of the increasing complexity of our modeling for reconstruction and accuracy on all datasets in both the supervised and unsupervised settings. Note that learning raw prototypes boils down to the NCC method (Cover & Hart, 1967) in the supervised setting and to the K-means algorithm (MacQueen, 1967) in the unsupervised setting.

| | | Supervised | | | | | | Unsupervised | | | | | |
|---|---|---|---|---|---|---|---|---|---|---|---|---|---|
| | | Val | | | Test | | | Train | | | Test | | |
| | | OA↑ | MA↑ | $\mathcal{L}_{\text{rec}}$ ↓ | OA↑ | MA↑ | $\mathcal{L}_{\text{rec}}$ ↓ | OA↑ | MA↑ | $\mathcal{L}_{\text{rec}}$ ↓ | OA↑ | MA↑ | $\mathcal{L}_{\text{rec}}$ ↓ |
| PASTIS | Raw prototypes | 57.3 | 50.0 | 4.43 | 56.5 | 48.4 | 4.46 | 69.1 | 29.8 | 2.77 | 69.0 | 29.8 | 2.78 |
| | + time warping | 56.8 | 53.7 | 4.00 | 56.2 | 51.4 | 4.04 | **69.2** | 30.4 | 2.53 | **69.1** | **30.4** | 2.53 |
| | + offset | 55.0 | 55.7 | **2.57** | 53.5 | 53.8 | **2.65** | 67.8 | 28.5 | **1.91** | 67.7 | 28.6 | **1.91** |
| | + $\mathcal{L}_{\text{cont}}$ | **74.8** | **61.3** | 2.90 | **73.7** | **59.1** | 3.00 | — | — | — | — | — | — |
| TS2C | Raw prototypes | 57.4 | 51.2 | 4.89 | 57.4 | 49.5 | 4.36 | 56.2 | 34.2 | 3.52 | 49.5 | 32.5 | 3.56 |
| | + time warping | 56.0 | 51.2 | 4.64 | 59.9 | 52.3 | 4.15 | 59.1 | 38.6 | 3.04 | **52.3** | **36.0** | 3.09 |
| | + offset | 56.9 | 51.8 | 3.50 | 57.3 | 55.0 | 3.49 | **60.0** | **39.3** | **2.40** | 52.0 | 35.5 | **2.53** |
| | + $\mathcal{L}_{\text{cont}}$ | **74.5** | **64.4** | 3.46 | **78.5** | **70.5** | 3.46 | — | — | — | — | — | — |
| SA | Raw prototypes | 54.8 | 50.0 | 3.43 | 51.3 | 46.4 | 4.62 | 60.9 | 50.9 | 1.43 | 61.9 | 47.8 | 1.85 |
| | + time warping | 57.5 | 53.9 | 2.93 | 54.5 | 49.7 | 4.13 | 62.2 | 53.1 | 1.03 | **64.1** | **51.7** | 1.46 |
| | + offset | 63.5 | 58.0 | **1.34** | 60.6 | 50.0 | 2.01 | **63.7** | **54.5** | **0.67** | 63.6 | 50.4 | **0.91** |
| | + $\mathcal{L}_{\text{cont}}$ | **71.0** | **64.7** | 1.89 | **62.3** | **54.9** | 2.66 | — | — | — | — | — | — |
| DENETH. | Raw prototypes | 68.3 | 58.0 | 3.89 | 61.3 | 55.5 | 4.58 | 63.8 | 52.8 | 2.67 | 57.2 | 48.5 | 2.41 |
| | + time warping | 70.1 | 59.5 | 3.52 | **62.4** | 56.4 | 4.21 | 64.8 | 54.0 | 2.23 | 57.6 | 51.1 | 2.01 |
| | + offset | 77.3 | 64.9 | **2.39** | 59.8 | **62.9** | 3.55 | **66.2** | **56.3** | **1.70** | **58.5** | **52.6** | **1.56** |
| | + $\mathcal{L}_{\text{cont}}$ | **85.1** | **75.5** | 3.00 | 56.5 | 54.2 | 4.35 | — | — | — | — | — | — |

### 5.3.2 Visualizing prototypes

We show in Figure 9 our prototypes and how they are deformed to reconstruct a given input. For each class of the SA dataset, we show an input time series that has been correctly assigned to its corresponding prototype by our model trained with supervision but without $\mathcal{L}_{\text{cont}}$. We see that the inputs are best reconstructed by a prototype of their class. Looking at any of the columns, we see that prototypes of other classes can also be deformed to reconstruct a given input, but only to a certain extent. This confirms that the transformations considered are simple enough so that the reconstruction power of each prototype is limited, but powerful enough to allow the prototypes to adapt to their input.

Figure 10 shows the 32 prototypes learned by our unsupervised model on SA, grouped by assigned label. For each prototype, we show an example input sample whose best reconstruction is obtained using this particular prototype and the obtained corresponding reconstruction. We see that prototypes are not equally assigned to classes, with class *Canola* having 14 prototypes when class *Small Grain Gazing* only has 1. This is due to the high imbalance of the classes in the datasets and different intra-class variabilities. Inside a class, different prototypes account for intra-class variability beyond what our deformations can model.

### 5.4 Discussion

DTI-TS fails at classifying an input pixel time series when the prototype of a wrong crop type is able to better reconstruct it than the prototype of the true class. This may happen in three cases that we detail below: (i) because both classes are very similar, (ii) because our deformations are powerful enough to align semantically different prototypes to the same input sequence or (iii) simply because the input time series is a difficult sample to reconstruct.

**Similar classes.** Example of similar classes can be seen in Figure 4 where the mean NDVI over time of the Winter wheat and the Rye classes on TS2C as well as the Barley and Rye classes of DENETHOR are very

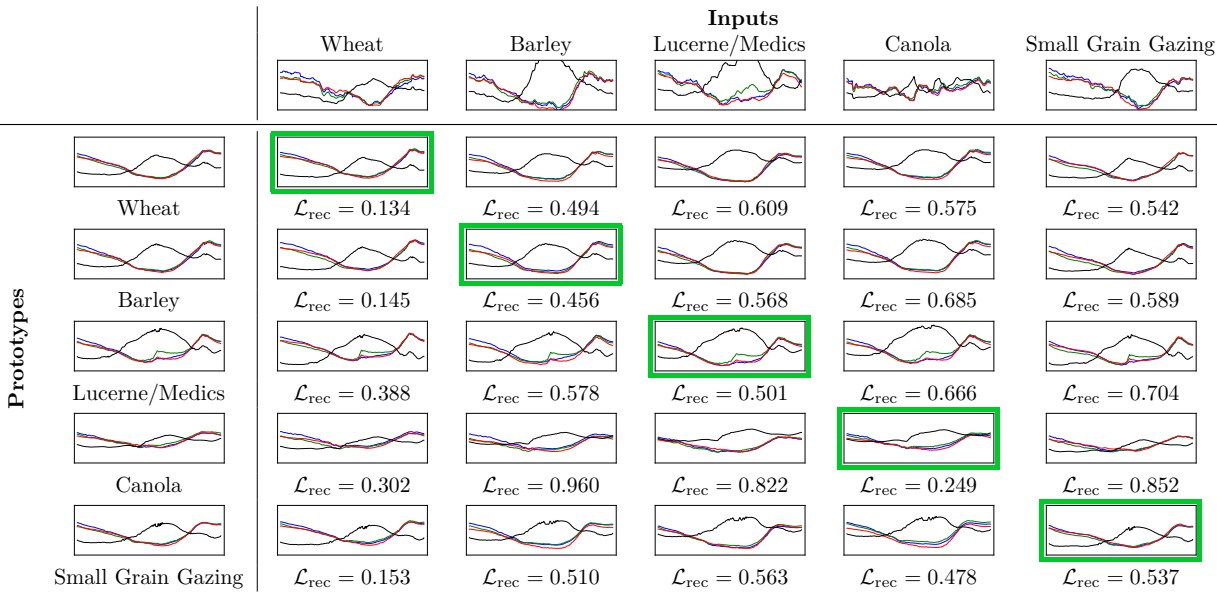

Figure 9: **Reconstructions from different prototypes.** We show the reconstructions of input samples (columns) from SA (Kondmann et al., 2022) by learned prototypes (rows) in the supervised setting without $\mathcal{L}_{\text{cont}}$. Selected prototypes (frames) correspond to the lowest reconstruction error.

Table 5: **Reconstruction loss of correct and wrong predictions.** We report the average reconstruction loss of correct and wrong predictions on all datasets for our method in the supervised case. We highlight in bold the lowest reconstruction loss for each row: time series that we tend to misclassify are also more difficult to reconstruct in general.

|  | ✓ Correct predictions | ✗ Wrong predictions |
|---|---|---|
| PASTIS (Garnot & Landrieu, 2021) | **2.52** | 2.72 |
| TS2C (Weikmann et al., 2021) | **3.44** | 3.79 |
| SA (Kondmann et al., 2022) | **1.84** | 2.26 |
| DENETHOR (Kondmann et al., 2021) | **3.54** | 3.57 |

close. Our transformations may align indifferently both class prototypes to an input sequence, discarding small differencies that would have helped classify it.

**Transformation design.** While our deformations are simple, they may not be constrained enough for the task of crop classification. The time warping stretches or squeezes temporally a time series using uniformly spaced control points. In Figure 4, looking at the Wheat class for SA, note how this time warping is able to align train (in blue) and test (in red) curves, despite a clear temporal shift. Even though these deformations are limited to 7 days in each direction, they do not focus on a specific period in the year. Our offset transformation assumes that intra-class spectral distortions are time-independent. Though we show empirically that we can reconstruct better time series when using this transformation, this comes at the price of reduced classification performance. We believe performances could be further improved by the design of physics-based transformations that could account for actual meteorologic events.

**Reconstruction performance on misclassified samples.** Misclassified samples by our method tend to be the most difficult to reconstruct. This statement is supported quantitatively in Table 5 where we show that the reconstruction loss is higher on average for misclassified time series on all datasets. In Figure 11, we can see that wrongly classified time series (in red) often show clear differences from the learned prototype

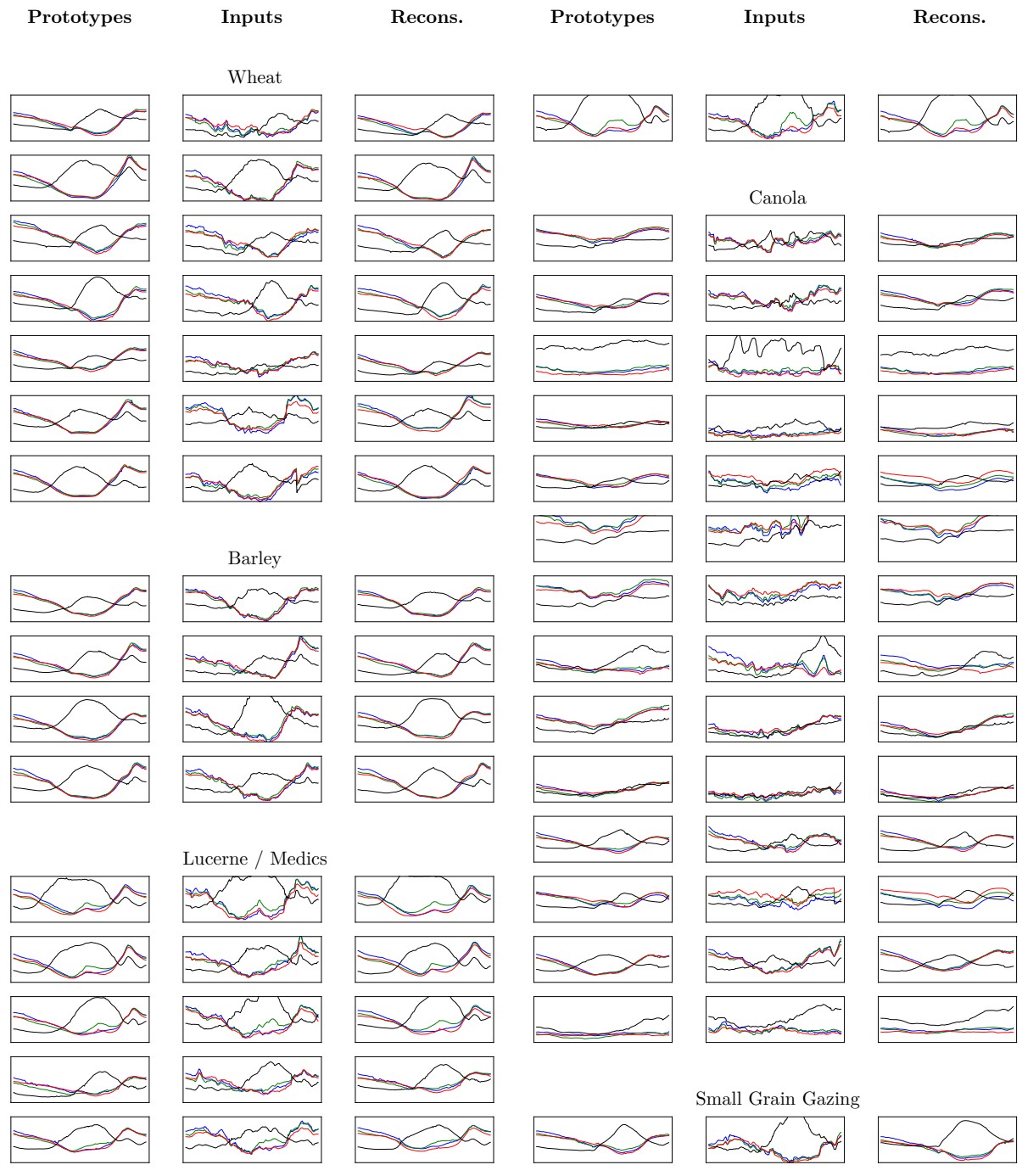

Figure 10: **Learned prototypes on SA.** We show the 32 prototypes learned on the SA dataset (Kondmann et al., 2022) (first column) in the unsupervised setting with time warping and offset deformations. For each prototype, we show an example time series of the corresponding class from the test set that is best reconstructed by it (second column) along with its reconstruction by our model (third column).

(in bold blue). Again, better suited deformations for this task should help prototypes reconstruct more accurately diverse time series of the same class and meanwhile not let them fit times series of other classes. We believe this to be a challenge that should be addressed in future work.

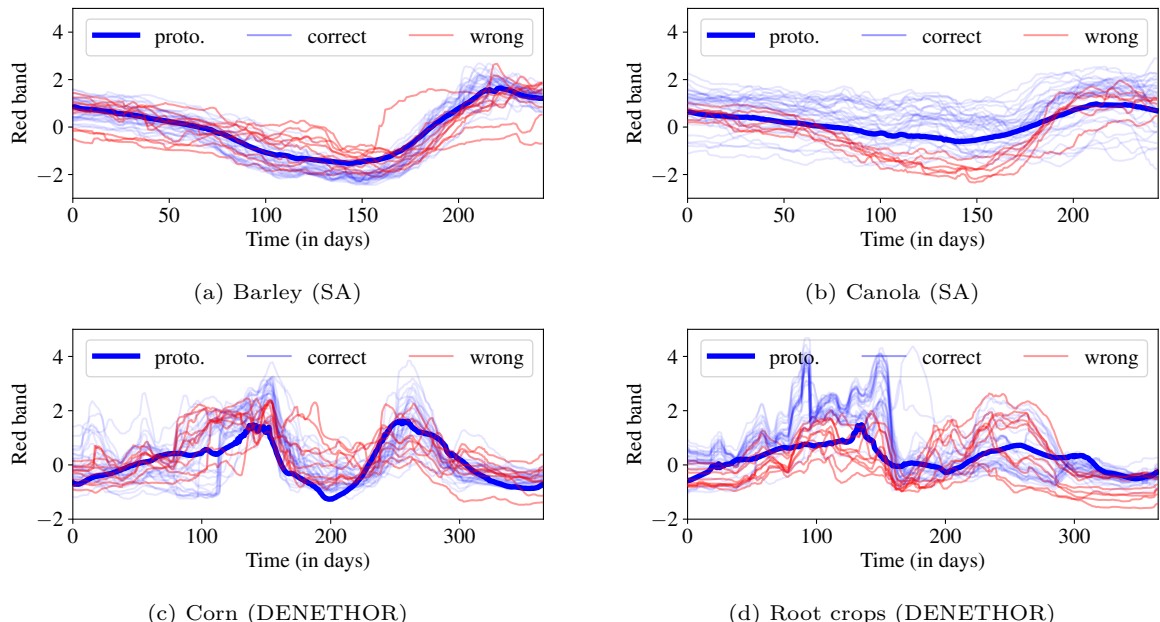

Figure 11: **Visual representation of failure cases.** We show the normalized red band of randomly selected time series from two classes of SA (Kondmann et al., 2022) and DENETHOR (Kondmann et al., 2021). We distinguish between time series correctly classified by our model (in blue) and time series misclassified (in red). We also display the red band of the corresponding learned prototype in each case.

## 6  Conclusion

We have presented an approach to learning invariance to transformations relevant for agricultural time series using deep learning, and demonstrated how it can be used to perform both supervised and unsupervised pixel-based classification of crop SITS. We perform our analysis on four recent public datasets with diverse characteristics and covering different countries. Our method significantly improves the performance of NCC and K-means on all datasets, while keeping their interpretability. We show it improves the state of the art on the DENETHOR dataset for classification. This result emphasizes the need for more multi-year datasets to reliably evaluate the potential of automatic methods for practical crop segmentation scenarios, for which our deformation modeling approach seems to provide significant advantages. DTI-TS also achieves best results in a low data regime on PASTIS, and on all datasets for unsupervised clustering. Additionally, we provide a benchmark of MTSC classification approaches for agricultural SITS classification.

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

## Appendix A - Prototype initialization

NCC or K-means centroids are used to initialize our prototypes, thus impacting the performance of our method. Plus, it is not obvious what is the best way to run NCC or K-means when faced with time series with missing data. The centroids can be computed by only giving weight to existing data points or after a gap filling operation. In this section, we focus on the supervised case and investigate other simple gap filling methods and the respective performance of both NCC and our method. Following the notations of Section 3.3, we define the following gap filling methods:

**None**   No gap filling is done and the data processed by the method correspond to the raw input data: $\mathbf{x} = \mathbf{x}_{\text{raw}}$ and $\mathbf{m} = \mathbf{m}_{\text{raw}}$.

**Previous**   Missing time stamps take the value of the closest previous data point in the time series:

$$\mathbf{x}[t] = \mathbf{x}_{\text{raw}}\Big[\max_{t' \leq t}\ \mathbf{m}_{\text{raw}}[t'] = 1\Big], \tag{A1}$$

and

$$\mathbf{m}[t] = \mathbb{1}_{\{t' < t | \mathbf{m}_{\text{raw}}[t']=1\} \neq \emptyset}[t]. \tag{A2}$$

**Moving average**   The value of each time stamps is set as the non-weighted average of the data points inside of a centered time window:

$$\mathbf{x}[t] = \frac{1}{\mathbf{m}[t]} \sum_{t'=t-\sigma}^{t+\sigma} \frac{\mathbf{x}_{\text{raw}}[t']}{2\sigma + 1}, \tag{A3}$$

where $\sigma$ is a hyperparameter set to 7 days in our experiments - same as in Equation 12. We also define the associated filtered mask $\mathbf{m}$ for $t \in [1, T]$ by:

$$\mathbf{m}[t] = \sum_{t'=t-\sigma}^{t+\sigma} \frac{\mathbf{m}_{\text{raw}}[t']}{2\sigma + 1}, \tag{A4}$$

for $t \in [1, T]$ and with the same hyperparameter $\sigma$.

**Gaussian filter**   We can consider the filtering of Equations 11 to 13 as a gap filling method.

The NCC centroid corresponding to class $k$ is then given by:

$$\mathbf{C}_k[t] = \frac{1}{N_k C} \sum_{\substack{i=1 \\ y_i = k}}^{N} \frac{\mathbf{m}_i[t]}{\sum_{t'=1}^{T} \mathbf{m}_i[t']} \mathbf{x}_i. \tag{A5}$$

In Table A1, we report the MA of both NCC and our method on TS2C dataset, using these different gap filling settings to compute NCC centroids. For our method, the only difference between experiments is the initialization of the prototypes. Filling missing data with Gaussian filtering improves over no gap filling by almost +5pt of MA. We also see that the ranking of the different gap filling methods is preserved with our method which confirms the importance of the initialization of the prototypes when training our model. Qualitatively, Figure A1 illustrates that our approach does not effectively address the inadequate quality of NCC centroids when gap filling is not employed. In contrast, all three gap filling strategies yield similar learned potato prototypes, even when initialized with NCC centroids that display substantial differences.

In Section 3.3, we also present a filtering scheme of input data to prevent learning from potential outliers. Note that this can also be done during the assignment step when running NCC. Table A1 also shows how this input filtering is necessary for both NCC and our method to reach their best performance.

Table A1: **Comparison of various initialization of NCC centroids** and resulting performance with our method. We also show the effect of applying a Gaussian filter on the input data following Equation 11.

| Gap filling | Input filtering | NCC | | DTI-TS | |
|---|---|---|---|---|---|
| | | OA | MA | OA | MA |
| None | ✗ | 53.1 | 45.3 | 69.2 | 58.5 |
| Previous | ✗ | 56.8 | 48.1 | 72.8 | 63.1 |
| Moving average | ✗ | 56.6 | 49.1 | 73.2 | 63.9 |
| Gaussian filter | ✗ | 57.1 | **49.9** | 76.4 | 68.2 |
| | ✓ | **57.7** | **49.9** | **78.5** | **70.5** |

(a) NCC+None      (b) DTI-TS+None

(c) NCC+Previous      (d) DTI-TS+Previous

(e) NCC+Moving average      (f) DTI-TS+Moving average

(g) NCC+Gaussian filter      (h) DTI-TS+Gaussian filter

Figure A1: **NCC centroids and learned prototypes. RGB** and **IR** spectral bands of centroids and prototypes obtained with NCC and our method respectively using different gap filling methods and corresponding to TS2C potato class.

## Appendix B - **Choice of** $K$

The number of prototypes $K$ under supervision exactly corresponds to the number of ground truth classes. Without ground truth labels, the number of prototypes is selected arbitrarily and should hopefully be higher than the number of expected true classes. In Figure B1 we report the MA of our method with and without offset for different numbers of learned prototypes on TS2C dataset. Being entirely unsupervised, there is no restriction on how prototypes relate to classes: complex classes can be represented by several prototypes and others only by a single prototypical time series as shown in Figure 10. The value $K = 32$ prototypes appears to provide a favorable balance between classification accuracy and the number of learned parameters. As a result, we conducted all our unsupervised experiments using this chosen value.

## Appendix C - **Prediction aggregation**

Pixel-wise methods in the scope of this paper do not leverage any spatial information or context. Thus, it is expected for whole-image based approaches like UTAE to reach better performance. However it is interesting to look for simple, yet effective fashions to aggregate pixel-wise predictions at the field level in a post-

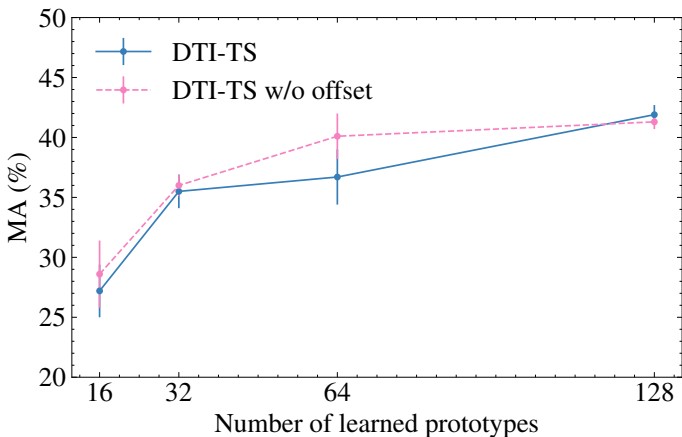

Figure B1: **Number of learned prototypes.** Mean accuracy of our method on TS2C depending on the number of learned prototypes. We show results averaged over 5 runs.

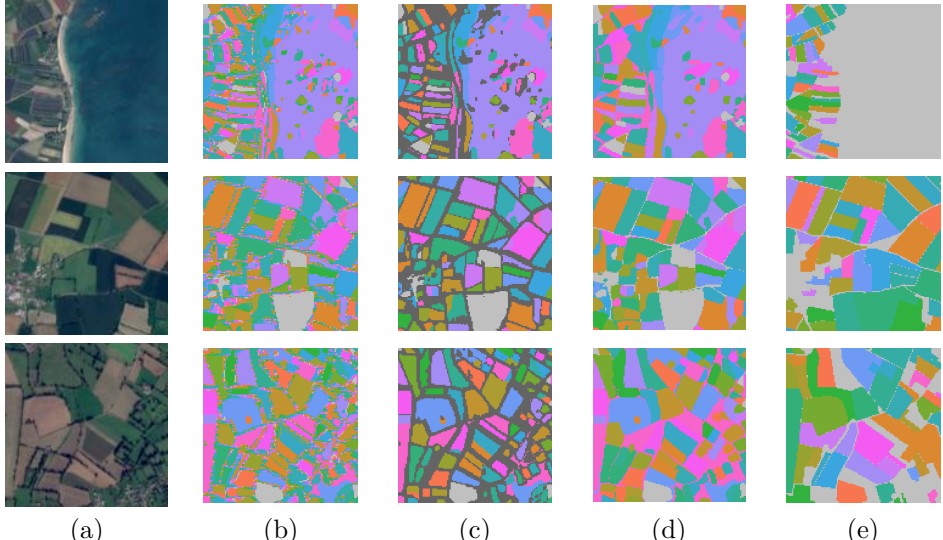

Figure C1: **SAM for SITS instance segmentation.** For randomly selected SITS from Fold 2 test set of PASTIS (a), we first show the fine segmentation $SAM_{raw}$ obtained when intersecting all temporal SAM outputs (b). Then we show the filtered maps $SAM_{filt}$ (c) where dark grey pixels correspond to remaining pixels. We show the final SAM-based instance maps (d) where all remaining pixels are assigned to one of the filtered instance and compare to the ground truth instance map (e).

processing step. In this section, we aggregate predictions using (i) ground truth instance segmentation maps, (ii) sliding windows or (iii) instance segmentation maps obtained with Segment Anything Model (Kirillov et al., 2023). The following study is performed on PASTIS Fold 2.

**Ground truth instance segmentation (GTI).** We can use the ground truth segmentation of agricultural parcels provided with PASTIS dataset to have an upper bound of what can be achieved in terms of field-level aggregation. For each given parcel, we use majority voting to assign to all pixels of the instance the corresponding label.

**Sliding patches (SW).** We assign to a pixel the majority label inside a patch of size 5×5 centered on it.

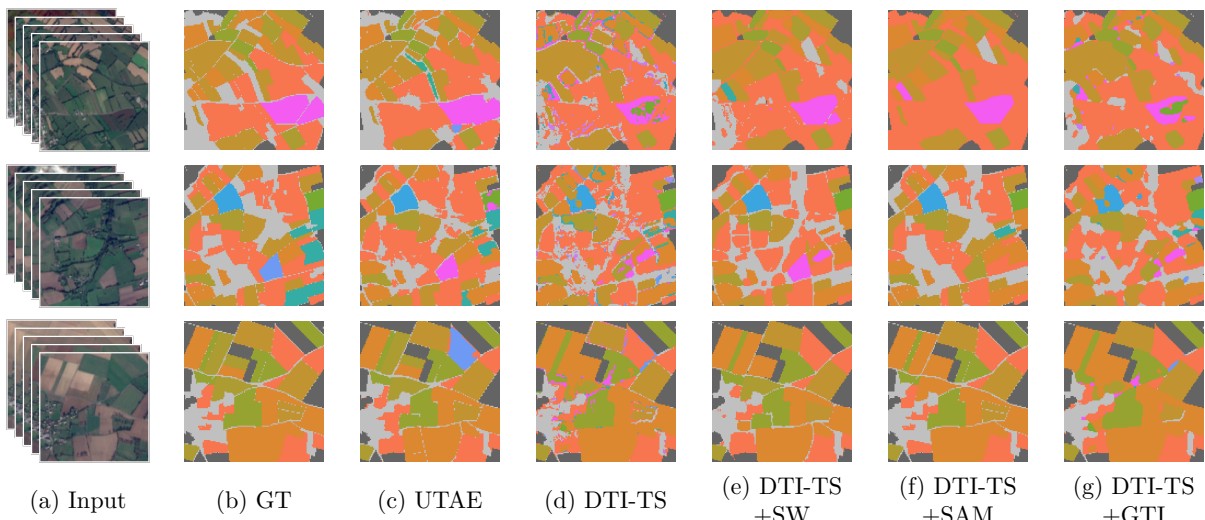

| (a) Input | (b) GT | (c) UTAE | (d) DTI-TS | (e) DTI-TS +SW | (f) DTI-TS +SAM | (g) DTI-TS +GTI |

Figure C2: **Qualitative comparisons of post-processing methods for field-level prediction aggregation.** We show predicted segmentation maps for UTAE image-based method (c) and compared it them to those obtained with our method with and without image-level post-processing methods (d-g) for randomly selected SITS from Fold 2 test set of PASTIS (a). Grey segments correspond to the *void* class and are ignored by all methods.

Table C1: **Quality of SAM instances.** We investigate performance of our method on all the pixels, on remaining pixels and on pixels in SAM filtered instances for Fold 2 test set of PASTIS.

|  | OA | MA |
| --- | --- | --- |
| All | 75.2 | 60.1 |
| Remaining pixels | 63.3 | 48.3 |
| Filtered instances | 80.4 | 66.7 |

**Segment Anything Model (SAM).** SAM (Kirillov et al., 2023) is an image segmentation model trained on 1 billion image masks. It is able to generate masks for an entire image or from a given prompt. Here, we use it off-the-shelf without any additional learning to generate instance segmentation maps for each image of a given SITS. Combining these possibly contradictory segmentation maps is not easy and is the subject of several related works (Franek et al., 2010; Li et al., 2012; Khelifi & Mignotte, 2016; Lefèvre et al., 2019). As in Lefèvre et al. (2019), we first produce a fine segmentation $\text{SAM}_{\text{raw}}$ by intersecting all the obtained maps: two pixels $p_1$ and $p_2$ belong in the same instance in the final result if and only if they belong in the same instance for all images of the time series *i.e.*:

$$d(p1, p2) = 0, \tag{C1}$$

with $d$ the number of images in the SITS where pixels $p_1$ and $p_2$ belong to different instances. Then we propose to only keep instances that are not empty when eroded with a $3{\times}3$ kernel. We distinguish the *filtered instances* from the *remaining pixels* on these $\text{SAM}_{\text{filt}}$ filtered instance segmentation maps. Examples of $\text{SAM}_{\text{raw}}$ and $\text{SAM}_{\text{filt}}$ maps can be found on Figure C1b and C1c respectively. In Table C1, we show that our method is more accurate on pixels of filtered instances than on remaining pixels, confirming that these instances correspond to clear and consistent spatial structures. Finally, a remaining pixel $p$ is assigned to the closest filtered instance containing a pixel $p'$ that minimizes $d(p, p')$. Example of final SAM-based instance maps are shown on C1d. We again use majority voting to assign to all pixels of an instance the corresponding label.

Table C2: **Bridging the gap between pixel-wise and whole-image methods.** We investigate how post-processing approaches can help our pixel-wise method be on par with a whole-image method like UTAE (Garnot & Landrieu, 2021).

| Method | Post-processing | OA | MA |
|--------|----------------|------|------|
| UTAE | None | 83.8 | 73.7 |
| DTI-TS | None | 75.2 | 60.1 |
| DTI-TS | SW | 74.8 | 61.3 |
| DTI-TS | SAM | 77.3 | 62.0 |
| DTI-TS | GTI | 86.4 | 67.5 |

We now compare quantitatively and qualitatively the prediction aggregation methods described above applied to our supervised method to the whole-image based approach UTAE (Garnot & Landrieu, 2021) on Fold 2 test set of PASTIS. On Figure C2, see how such post-processing steps allow to leverage spatial context in order to remove noisy predictions. While SW seems to rather smooth the raw semantic predictions than actually aggregating them at the field-level, SAM leads to results that are visually close to those obtained when using the ground-truth instances. Quantitatively though, we observe in Table C2 a neat gap between SAM and GTI (9.1% in OA and 5.5% in MA) which encourage to search for better instance proposing methods. Still, SAM post-processing lead to a +2% increase in both OA and MA, demonstrating that obtained segments are semantically consistent. Finally, using GTI, our approach outperforms UTAE by +2.6% in OA but is behind by -6.2% in MA. Here the post-processing especially help classify the majority classes.

