# OpenReview forum: "Pixel-wise Agricultural Image Time Series Classification: Comparisons and a Deformable Prototype-based Approach"
_TMLR — Withdrawn by Authors_

### Review · Reviewer_MUiz · 2024-04-18

**Summary Of Contributions:**

In this paper, they present and compare datasets and methods for both supervised and unsupervised pixel-wise segmentation of satellite image time series (SITS). They also introduce an approach to add invariance to spectral deformations and temporal shifts to classical prototype-based methods such as K-means and Nearest Centroid Classifier (NCC). Different levels of supervision and show this simple and highly interpretable method achieves the best performance in the low data regime and significantly improves the state
of the art for unsupervised classification of agricultural time series on four recent SITS datasets.

**Audience:**

Yes

**Broader Impact Concerns:**

No.

**Claims And Evidence:**

No

**Requested Changes:**

table 2 is not referenced by any sentence in this paper.

**Strengths And Weaknesses:**

Strengths:
They presented an approach to learning invariance to transformations relevant for agricultural time series using deep learning, and demonstrated how it can be used to perform both supervised and unsupervised pixel-based classification of crop SITS.
A transformation module corresponding to time warping which enables to adapt deep transformation-invariant (DTI) clustering (Monnier et al., 2020) to SITS classification and improve nearest centroid classifier (Cover & Hart, 1967).


Weaknesses and Questions:

Q1:
Table 2 indicates that the performance of the DTI-TS method significantly lags behind that of the state-of-the-art models across several datasets. Could the authors provide more insight into the potential reasons for this discrepancy?

Q2:
“Note however that in doing so, they lose the interpretability of simpler approaches such as 1NN or NCC, which our method is designed to keep”. In temporal attention mechanism (Garnot & Landrieu, 2020), they are indeed using the attention mechanism to capture the appropriate temporal pattern for each given prototype. While in another work (Tang et al., 2022), the multiple sized receptive fields is employed to help find the appropriate/accurate CNN window size/receptive field. Could the authors further elaborate on how the interpretability of 1NN, NCC, and DTI-TS compares, especially in the context of these developments?

Q3:
Considering that DTI-TS leverages spectro-temporal invariance, would the integration of data spectro-temporal invariance techniques alongside advanced model architectures such as OS-CNN or TapNet potentially enhance performance.

Q4:
The paper describes a transformation model that includes spectral offset and time warping, alongside a convolutional network architecture ending in a linear layer with K × (C + M) outputs activated by a tanh function. Are these architectural components applied consistently across both supervised and unsupervised tasks?

Q5:
Analysis of Table 4 suggests that in supervised tasks, the observed performance gains primarily arise from the introduction of contrastive loss, rather than the transformation model. Given the longstanding nature of k-means in unsupervised learning, have the authors considered enhancing their method with contemporary variants like K-means

---

### Review · Reviewer_S36E · 2024-05-15

**Summary Of Contributions:**

The authors introduce a Multivariate Time Series Classification approach to identify crop NDVI templates from remote sensing data.  The unsupervised settings enables the discovery of templates and the supervised setting enables crop type classification.  The performance is compared to 8 related methods across 4 different datasets.  Performance is comparable to or slightly better than other MTSC methods for crop type classification.  Code is made available.

**Audience:**

No

**Broader Impact Concerns:**

There are no concerns around broader impacts.

**Claims And Evidence:**

No

**Requested Changes:**

A confusion matrix would be beneficial.

Most crop-classification approaches today leverage the geospatial information, not just the spectral information as in this approach.  Obviously the focus of this work is on MTSC approaches, but these other methods are considered SOTA.  Especially with the advances of SSL methods to train on even more plentiful data, these approaches perform quite well.  Additional reference to these methods and why MTSC is preferred over using the geospatial information should be discussed.

Some discussion of how the templates vary between sensors and geographies would be beneficial.  Essentially, it would be good to quantify the stability of the approach to out of domain observations.

The discussion could be improved.  It immediately dives into referencing a failure case.  My suggestion would be to build into this and not just focus around limitations (although that can be a large segment of it).  Speaking to the strengths and advantages as well as what was learned

Related to that, the discussion is largely a list of limitations.  That's fine, but I would introduce that in the prelude to those elements.  Additionally, I believe a few limitations should be further discussed. In particular, the authors should speak to unseen/out of domain drops, multi-crop fields (which is becoming much more prevalent), performance as a function of plot-size, performance as a function of re-visit frequency (and missing visits), and "time in the season classification" (at what point can the crop be classified)?

I believe Figure 10 can be condensed and/or moved to the appendix.  Seeing all of those samples isn't particularly more informative that a few samples wouldn't highlight (and then the reader could look to the appendix if desired).

The authors discuss how missing data is handled in section 3.  However, I think additional experiments showing how the method performs as more and more data is lost is critical.  Sentinel, in particular, may have very long gaps in coverage because of clouds or shadows.  How long can a gap be before performance degrades?  How does this compare to other methods?

**Strengths And Weaknesses:**

## Strengths:
The methodology is straightforward and addresses the alignment both spatially and temporally.  Both the visual and mathematical descriptions are clear.

The NVDI trace clusters found are consistent with what is in the crop science literature.

Performance is tested on a variety of datasets from different sensors and over different locations.  Performance compared to other MTSC methods is strong.

The grammar and syntax throughout is clear.  Figures are also clear as well.

---
## Weaknesses:
Acceptance criteria is that of "interest" to the TMLR audience not "novelty" or "impact".  With that in mind, I still struggle to see how this would be of interest to the community because the methodology does not solve a meaningful **technical** problem.  There are many approaches to performing crop classification from remote sensing, and the proposed approach is not meaningfully better or more interesting than other approaches out there; it does not unlock a solution or solve a challenge that isn't already addressable by other means, nor is there particular insight which could be used to solve other problems.

To that end, it feels more like an "analysis" paper where templates of crop signatures are found, and could be used for classification if desired.  Are those templates useful?  That's unclear.  While they **can** be used for crop classification, other approaches give comparable or better results.  Are they useful in revealing something about crop health, yield, or something else?  That wasn't explored here.  The discovery of clusters in NDVI trends is not something I would expect the TMLR community to find interesting (perhaps the crop science community) and the methodology performance is not unique or performant enough that I would see the community interested in that regard.  The paper feels more like a demonstration of largely traditional timeseries approaches on remote sensing data because it's a nice visual demonstration, but doesn't solve a problem or advance our understanding of time series analysis or crop type classification.

The authors state that their contribution is a benchmarking of MTSC approaches on four datasets.  At the strictest level, yes, this is benchmarking, but these would be required anyway to compare the performance of their new approach.  So I don't see that as an additional contribution beyond the introduction of this new approach.  A true benchmarking study would look at a wider range of approaches (including those using the spatial information) and explore under what scenarios each performs better/worse (in vs. out of domain, different sensors, stability against missing data, ability to handle multi-cropping, etc.) along different metrics (accuracy metrics, point in the season at which prediction can be made, inference speed, training time, etc.).   Selecting a handful (even 8) of related methods for comparison is not a benchmarking study- in fact, benchmarking studies should look at widely different approaches, not very similar ones.

A major omission and/or limitation of this work is the need to see the full template.  Is the full template, i.e. season required in order to classify the crop?  If so this would dramatically limit its applicability.  If not, how does performance vary as a function of time since planting/emergence (overall as well as a function of crop)?  Similarly, how is transition between crops handled when the start is not known?  For example, if a plot has corn one year, a cover crop in winter, and then soy the next, but the "start" of season is not known, how will this be handled?  There are a lot of practical considerations which need to be discussed (both additional results as well as discussion) to explain how this would be useful in the real-world setting, as this is primarily an "application" paper, focused on a very narrow use case (i.e. crop classification from remote sensing).

Additional exploration around missing data is needed (see requested changes).

---

### Review · Reviewer_6HRS · 2024-05-26

**Summary Of Contributions:**

The paper presents a segmentation method for satellite image time series (SITS) under supervised and unsupervised learning settings. In particular, this work formulates the segmentation problem as a pixel-sequence classification/clustering task. To tackle the tasks, it proposes a prototype learning strategy based on the deep transformation invariance framework, and defines hybrid losses aiming to produce a prototype for each class with invariance to transformations including magnitude shift and temporal warping. The proposed method is evaluated on four datasets with comparisons to multiple time series classification and clustering baselines.

**Audience:**

Yes

**Broader Impact Concerns:**

No broader impact concerns are presented.

**Claims And Evidence:**

No

**Requested Changes:**

Please address the weaknesses mentioned above, which are important for my recommendation.

**Strengths And Weaknesses:**

Pros:
1. The paper presents an alternative strategy to SITS based on the time-series analysis clustering technique, which seems new in the context of this application. Given such a formulation, it benchmarked several multivariate time series classification methods, showing their limitations under realistic settings such as DNETHOR.

2. It develops a variant of DTI clustering framework for SITS, which can be applied in supervised or unsupervised cases. Empirically, it demonstrates a competitive performance under certain datasets and settings.

3. The paper is well-written and easy to follow.

Cons:
1. The motivation of the proposed formulation is not well justified. While it might be interesting to see how an alternative technique performs for this specific task, pixel-wise classification seems unsuitable for semantic segmentation as it requires spatial context to make reasonable predictions. In particular, satellite images are typically noisy and with missing data, which makes contextual cues more valuable. Recent work (ViTs for SITS) shows much better performance on those benchmarks.

2. There are several time-series alignment works after DTAN, which are not cited and compared.

      - Regularization-free Diffeomorphic Temporal Alignment Nets, ICML 2023

      - Closed-form diffeomorphic transformations for time series alignment, ICML 2022

3. The paper seems missing the literature on self-supervised learning for the time series. It is unclear how well the proposed method performs against using the representation learned by those SS strategies plus classifier finetuning or clustering.

     - TempCLR: Temporal Alignment Representation with Contrastive Learning, ICLR 2023

     - T-Rep: Representation Learning for Time Series using Time-Embeddings, ICLR 2024

4. The proposed loss seems a bit restrictive as it assumes all the clusters/classes share the same variance (as Kmeans does). Moreover, its supervised loss seems slightly ad hoc, requiring a combination of terms from different prior methods. What about using GMM for the unsupervised case and a standard classification loss for the supervised case?

5. The experimental evaluation is a bit lacking in the following aspects.
   - For the evaluation metric, it should report mIOU as this is a segmentation task.
   - The overall performance of the proposed method is mixed and its top performance requires using different loss settings (see Table 2 and 3). This casts doubts on how effective this framework is.
   - The ablation study shows that several proposed components can hurt the test performance (Table 4), which also makes their contribution unclear.

---

### Note · Authors · 2024-05-29

**Comment:**

We thank the reviewers for their insightful feedback and understand their recommendations. However, we deem the required modifications of the paper too significant to be treated within the TMLR review process and choose to withdraw the paper.

**Withdrawal Confirmation:**

I have read and agree with the venue's withdrawal policy on behalf of myself and my co-authors.